# Some New Constructions of *q*-ary Codes for Correcting a Burst of at Most *t* Deletions [note 1]

**DOI:** 10.3390/e27010085

**Published:** 2025-01-18

**Authors:** Wentu Song, Kui Cai, Tony Q. S. Quek

**Affiliations:** Science, Mathematics and Technology (SMT) Cluster, Singapore University of Technology and Design, Singapore 487372, Singapore; wentu_song@sutd.edu.sg (W.S.); tonyquek@sutd.edu.sg (T.Q.S.Q.)

**Keywords:** deletion correcting codes, sequence reconstruction, reconstruction codes, burst-deletion

## Abstract

In this paper, we construct *q*-ary codes for correcting a burst of at most *t* deletions, where t,q≥2 are arbitrarily fixed positive integers. We consider two scenarios of error correction: the classical error correcting codes, which recover each codeword from one read (channel output), and the reconstruction codes, which allow to recover each codeword from multiple channel reads. For the first scenario, our construction has redundancy logn+8loglogn+o(loglogn) bits, encoding complexity $O(q^{7t}n{\left(\log n\right)}^{3})$ and decoding complexity O(nlogn). For the reconstruction scenario, our construction can recover the codewords with two reads and has redundancy 8loglogn+o(loglogn) bits. The encoding complexity of this construction is $O\left(q^{7t}n{\left(\log n\right)}^{3}\right)$, and decoding complexity is $O\left(q^{9t}{\left(n\log n\right)}^{3}\right)$. Both of our constructions have lower redundancy than the best known existing works. We also give explicit encoding functions for both constructions that are simpler than previous works.

## 1. Introduction

The study of deletion/insertion correcting codes, which originated in the 1960s, has made a great progress in recent years, encouraged by their application to DNA-based data storage. One of the basic problems in this area is construction of codes with low redundancy and low encoding/decoding complexity, where the redundancy of a *q*-ary (q≥2) code C of length *n* is defined as $n-\log_{q}\left|C\right|$, measured in *q*-ary symbols, or $(n-\log_{q}|C|)\log q$, measured in bits (in this paper, for simplicity, for any real x>0, we write $\log_{2}x=\log x$). The optimal redundancy of a *q*-ary *t*-deletion correcting code of length *n* was proved to be asymptotically between tlogn+tlogq+o(logqn) and 2tlogn+tlogq+o(logqn) [1]. In general, codes with a redundancy matching the upper bound can be constructed by graph-coloring method. However, the encoding complexity of such a construction is exponential in *n*. In practice, the construction of codes with polynomial encoding complexity (also called explicit construction) and low redundancy is an interesting research problem.

The famous VT codes were proved to be a family of single-deletion correcting binary codes, and are asymptotically optimal in redundancy [2]. The VT construction was generalized to nonbinary single-deletion correcting codes in [3] and, recently, a different version of nonbinary VT codes was proposed in [4] using differential vector, with asymptotically optimal redundancy and efficient encoding/decoding. Works on binary and nonbinary codes for correcting multiple deletions can be found in [5,6,7,8,9,10,11,12,13] and the references therein.

Burst deletions and insertions, which means that deletions and insertions occur at consecutive positions in a string, are a class of error that can be found in many applications, such as DNA-based data storage and file synchronization. For the binary case, the maximal cardinality of a *t*-burst-deletion correcting code (i.e., a code that can correct a burst of *exactly t* deletions) is proved to be asymptotically upper bounded by $2^{n-t+1}/n$ [14], so its redundancy is asymptotically lower bounded by logn+t−1. Several constructions of binary codes correcting a burst of exactly *t* deletions were reported in [15,16], where the construction in [16] achieves an optimal redundancy of logn+(t−1)loglogn+k−logk. A more general class, i.e., codes correcting a burst of *at most t* deletions, were also constructed in the same paper [16], and this construction was improved in [17] to achieve a redundancy of $\left\lceil \log t\right\rceil \log n+\left(t\left(t+1\right)/2-1\right)\log\log n+c_{t}$ for some constant $c_{t}$ that only depends on *t*. In [18], by using VT constraint and shifted VT constraint in the so-called (p,δ)-dense strings, binary codes correcting a burst of at most *t* deletions were constructed, with an optimal redundancy of $\log n+t\left(t+1\right)/2\log\log n+c_{t}^{\prime }$, where $c_{t}^{\prime }$ is a constant depending only on *t*.

In recent parallel works [12,19], *q*-ary codes correcting a burst of at most *t* deletions were constructed for even integer q>2, with redundancy logn+O(logqloglogn) or, more specifically, $\log n+\left(8\log q+9\right)\log\log n+\gamma_{t}^{\prime }+o\left(\log\log n\right)$ bits for some constant $\gamma_{t}^{\prime }$ that only depends on *t*. The basic techniques in [12,19] are to represent each *q*-ary string as a binary matrix whose columns are the binary representation of the entries of the corresponding *q*-ary string. Strings of length *n* are constructed such that the first row of their matrix representation is (p,δ)-dense. Then, the first row of their matrix representation is protected by a binary burst deletion correcting code of length *n* and the other rows are protected by binary burst deletion correcting codes of length not greater than 2δ, which results in a construction with logn+O(logqloglogn) bits of redundancy. A different construction of *q*-ary codes correcting a burst of at most *t* deletions was reported in a more recent work [4], which has redundancy $\log n+O\left(t^{2}\log\log n\right)+O\left(t\log q\right)$.

A relaxed model of error correction, called sequence reconstruction, also received great attention from researchers (e.g., see [20,21,22,23,24,25,26,27,28,29,30,31,32]). Unlike the classical error correcting codes, in the sequence reconstruction model, the receiver is allowed to reconstruct the original transmitted sequence from multiple noisy reads (channel outputs). This model is suitable for DNA data storage because current synthesis and sequencing technologies can generate many (possibly erroneous) reads for each DNA strand, and so each stored DNA strand can be recovered by its many erroneous copies. Sequence reconstruction for deletion, insertion, transposition, and substitution was first studied in [20,21], where the minimum number of reads for exact reconstruction of uncoded sequence was computed. Coded sequence reconstruction for deletion channel was considered recently in [23], where it was assumed that a codeword of a single-deletion-correcting code is transmitted over the *t*-deletion channels, and the minimum number of distinct reads required to uniquely reconstruct the transmitted sequence was computed. The more general problem, i.e., the minimum number of reads for reconstruction of a codeword of an (ℓ−1)-deletion-correcting code of length *n* transmitted over the *t*-deletion channels for some 1≤ℓ≤t<n, was solved in [27]. The dual problem, i.e., designing codes (called reconstruction codes) for reconstruction of a sequence with fixed number of reads for deletion channel, was also considered in recent years. A construction of binary reconstruction codes for two reads and with loglogn+O(1) bits of redundancy under single-deletion channel was presented in [24], and this construction was generalized in [28] to *q*-ary single-edit channel for q≥2. Binary reconstruction codes under 2-deletion channel were constructed in [29,30]. It was shown in [30] that 3logn+o(logn) bits of redundancy is sufficient for two reads, and logn+o(logn) bits of redundancy is sufficient for five reads. Reconstruction codes under single-burst-insertion/deletion/edit channel were considered in [31], where for the channel suffering from a single burst of at most *t* deletions, a family of *q*-ary codes for two reads with t(t+1)/2loglogn+O(1) bits of redundancy were constructed.

In this paper, we propose some new constructions of *q*-ary codes for correcting a burst of at most *t* deletions for any fixed positive integers *t* and q≥2. We consider both the classical error correcting codes and the reconstruction codes. In our constructions, we consider *q*-ary (p,δ)-dense strings (sequences), which are generalization of the binary (p,δ)-dense strings defined in [18], and give an efficient algorithm for encoding and decoding of *q*-ary (p,δ)-dense strings. For the classical burst-deletion correcting codes, a VT-like function is used to locate the deletions within an interval of length not greater than 3δ, which results in logn bits of redundancy. In addition, two functions are used to recover the substring destroyed by deletions, which results in 8loglogn+o(loglogn) bits of redundancy (in this paper, the term o(loglogn) may depends on *q* and *t*. However, since *q* and *t* are assumed to be fixed positive integers, they are omitted). Thus, the total redundancy of our construction is logn+8loglogn+o(loglogn) bits. Compared to previous works, the redundancy of our new construction is independent of *q* and *t* in the second term. An explicit encoding function is given, which is simpler than previous works and has complexity $O(q^{7t}n{\left(\log n\right)}^{3})$. The decoding complexity is O(nlogn).

We also construct reconstruction codes for correcting a burst of at most *t* deletions from two reads and with redundancy 8loglogn+o(loglogn) bits, which is lower than the construction in [31]. We give an explicit encoding function for such codes with encoding complexity $O\left(q^{7t}n{\left(\log n\right)}^{3}\right)$ and decoding complexity $O\left(q^{9t}{\left(n\log n\right)}^{3}\right)$.

In Section 2, we introduce related notations and concepts, and present some basic constructions that will be used in our new constructions. In Section 3, we study (p,δ)-dense *q*-ary strings. A new construction of classical *q*-ary burst-deletion correcting codes is given in Section 4, and *q*-ary codes for correcting burst deletions from two reads is given in Section 5. Finally, the paper is concluded in Section 6.

## 2. Preliminaries

Let [m,n]={m,m+1,…,n} for any two integers *m* and *n*, such that m≤n, and call [m,n] an *interval*. If m>n, then let [m,n]=Ø. For simplicity, we denote [n]=[1,n] for any positive integer *n*. The size of a set *S* is denoted by |S|.

Given any integer q≥2, let $\Sigma_{q}=\left\{0,1,2,\cdots ,q-1\right\}$. For any sequence (also called a string or a vector) $x\in \Sigma_{q}^{n}$, *n* is called the length of x, and denote |x|=n. We will denote $x=(x_{1},x_{2},\ldots ,x_{n})$ or $x=x_{1}x_{2}\cdots x_{n}$. For any set $I=\left\{i_{1},i_{2},\ldots ,i_{m}\right\}\subseteq \left[n\right]$ such that $i_{1}<i_{2}<\cdots <i_{m}$, denote $x_{I}=x_{i_{1}}x_{i_{2}}\cdots x_{i_{m}}$, and call $x_{I}$ a *subsequence* of x. If I=[i,j] for some i,j∈[1,n] such that i≤j, then $x_{I}=x_{\left[i,j\right]}=x_{i}x_{i+1}\cdots x_{j}$ is called a *substring* of x. We say that x
*contains*
p( or p is contained in x) if p is a substring of x. For any $x\in \Sigma_{q}^{n}$ and $y\in \Sigma_{q}^{n^{\prime }}$, we use xy to denote their *concatenation*, i.e., $xy=x_{1}x_{2}\cdots x_{n}y_{1}y_{2}\cdots y_{n^{\prime }}$. We also use notations such as $x_{0},x_{1},\cdots ,x_{k}$ to denote substrings of a sequence x. For example, the notation $x=x_{1}x_{2}\cdots x_{k}$ means that the sequence x consists of *k* substrings $x_{1},x_{2},\cdots ,x_{k}$.

Let t≤n be a nonnegative integer. For any $x\in \Sigma_{q}^{n}$, let $D_{t}\left(x\right)$ denote the set of subsequences of x of length n−t, and let $B_{t}\left(x\right)$ denote the set of subsequences y of x that can be obtained from x by a burst of *t* deletions, that is, $y=x_{[n]\setminus D}$ for some interval D⊆[n] of length t (i.e., D=[i,i+t−1] for some i∈[n−t+1]). Moreover, let $B_{\leq t}\left(x\right)={⋃}_{t^{\prime }=0}^{t}B_{t^{\prime }}\left(x\right)$, i.e., $B_{\leq t}\left(x\right)$ is the set of subsequences of x that can be obtained from x by a burst of at most *t* deletions. Clearly, $B_{1}\left(x\right)=D_{1}\left(x\right)$ and $B_{t}\left(x\right)\subseteq D_{t}\left(x\right)$ for t≥2.

A code $C\subseteq \Sigma_{q}^{n}$ is said to be a *t*-*deletion correcting code* if, for any codeword x∈C, given any $y\in D_{t}\left(x\right)$, x can be uniquely recovered from y; the code $C\subseteq \Sigma_{q}^{n}$ is said to be capable of *correcting a burst of at most t deletions* if, for any x∈C, given any $y\in B_{\leq t}\left(x\right)$, x can be uniquely recovered from y. More generally, let $B\in \{D_{t},B_{\leq t}\}$ and *N* be a positive integer. A code $C\subseteq \Sigma_{q}^{n}$ is said to be an (n,N,B)-*reconstruction code* if, for any codeword x∈C, x can be uniquely recovered from any given *N* distinct sequences in B(x). In this case, we also say that x can be uniquely recovered from *N* reads in B(x). If N=1, then an (n,N,B)-reconstruction code degenerates to the classical error correcting code for the error pattern B.

For any code $C\subseteq \Sigma_{q}^{n}$, the redundancy of C is defined as $n-\log_{q}\left|C\right|$ measured in *q*-ary symbols or $\log q(n-\log_{q}|C|)$ measured in bits. Clearly, if there is an encoding function that maps each length-*k* sequence (message) to a length-*n* codeword in C, then the redundancy of C is n−k. In this paper, we will always assume that *q* and *t* are fixed (i.e., *q* and *t* are constant with respect to n).

A convenient way for constructing deletion correcting codes is to construct some sketches such that for sufficiently many sequences, each can be recovered from its (known) sketches and one of its subsequence obtained by (a burst of) at most *t* deletions. The VT syndrome is a sketch for correcting a single deletion, which is defined as follows. For each $c=\left(c_{1},c_{2},\cdots ,c_{n}\right)\in \Sigma_{2}^{n}$, the VT syndrome of c is defined as$$\mathrm{VT}\left(c\right)=\sum_{i=1}^{n}ic_{i}\;\;\mathrm{mod}\;\left(n+1\right).$$

It was proved in [2] that for any $c\in \Sigma_{2}^{n}$, given VT(c) and any $y\in D_{1}\left(c\right)$, one can uniquely recover x.

Suppose q>2. For each $x=\left(x_{1},x_{2},\cdots ,x_{n}\right)\in \Sigma_{q}^{n}$, let $\phi \left(x\right)=\left(\phi {\left(x\right)}_{1},\phi {\left(x\right)}_{2},\cdots ,\phi {\left(x\right)}_{n}\right)\in \Sigma_{2}^{n}$ be such that, for each i∈[2,n], $\phi {\left(x\right)}_{i}=1$ if $x_{i}\geq x_{i-1}$ and $\phi {\left(x\right)}_{i}=0$ if $x_{i}<x_{i-1}$. Moreover, let $\phi {\left(x\right)}_{1}=0$ for all $x\in \Sigma_{q}^{n}.\;($ one can also let $\phi {\left(x\right)}_{1}=1$ for all $x\in \Sigma_{q}^{n})$. Then, there are *q*-ary codes constructed in [3] for correcting a single deletion.

**Lemma** **1.**
*[3] For any $x\in \Sigma_{q}^{n}$, given $\mathrm{VT}(\phi {\left(x\right)}_{\left[2,n\right]})$, Sum(x) and any $y\in D_{1}\left(x\right)$, one can uniquely recover x, where $\phi {\left(x\right)}_{\left[2,n\right]}=\left(\phi {\left(x\right)}_{2},\cdots ,\phi {\left(x\right)}_{n}\right)$ and*

$$\mathrm{Sum}\left(x\right)=\sum_{i=1}^{n}x_{i}\;\mathrm{mod}\;q.$$



### 2.1. A Construction of *q*-ary Burst-Deletion Codes

For codes correcting burst deletions, the following lemma gives a *q*-ary version the construction in [33] to *q*-ary codes (q>2), and will be used in our new construction.

**Lemma** **2.**
*For any positive integer m, there exists a function*

$$h_{\mathrm{syn}}:\;\Sigma_{q}^{m}\to \Sigma_{q}^{4\log_{q}m+o\left(\log_{q}m\right)},$$

*computable in time $O(q^{t}m^{3})$, such that $h_{\mathrm{syn}}\left(x\right)\neq h_{\mathrm{syn}}\left(x^{\prime }\right)$ for any distinct $x,x^{\prime }\in \Sigma_{q}^{m}$ with $B_{\leq t}\left(x\right)\cap B_{\leq t}\left(x^{\prime }\right)\neq Ø$.*


**Proof.** The function $h_{\mathrm{syn}}$ can be constructed by the syndrome compression technique developed in [33].For each $x\in \Sigma_{q}^{m}$, let $N_{t}\left(x\right)$ be the set of all $x^{\prime }\in \Sigma_{q}^{m}\setminus \left\{x\right\}$, such that $B_{\leq t}\left(x\right)\cap B_{\leq t}\left(x^{\prime }\right)\neq Ø$. By simple counting, we have(1)$$\begin{array}{c} |N_{t}\left(x\right)|\leq tm^{2}q^{t}. \end{array}$$We first construct a function $\bar{h}:\Sigma_{q}^{m}\to \left[0,2^{\bar{R}}-1\right]$ satisfying: 1) $\bar{R}=\frac{t(t+1)}{2}\left(\log m+\log q\right)$; and 2) $\bar{h}\left(x\right)\neq \bar{h}\left(x^{\prime }\right)$ for any $x\in \Sigma_{q}^{m}$ and any $x^{\prime }\in N_{t}\left(x\right)$. Specifically, $\bar{h}$ is constructed as follows: For each $t^{\prime }\in \left[t\right]$ and $j\in [t^{\prime }]$, let$${\bar{h}}_{t^{\prime },j}\left(x\right)=\left(\mathrm{VT}\left(\phi {\left(x_{I_{t^{\prime },j}}\right)}_{\left[2,m_{t^{\prime },j}\right]}\right),\mathrm{Sum}\left(x_{I_{t^{\prime },j}}\right)\right),$$
where $I_{t^{\prime },j}=\left\{\ell \in \left[m\right]:\ell \equiv j\;\mathrm{mod}\;t^{\prime }\right\}$ and $m_{t^{\prime },j}=\left|I_{t^{\prime },j}\right|$. Then, let$$\bar{h}=\left({\bar{h}}_{1,1},{\bar{h}}_{2,1},{\bar{h}}_{2,2},\cdots ,{\bar{h}}_{t,1},{\bar{h}}_{t,2},\cdots ,{\bar{h}}_{t,t}\right).$$Clearly, $|I_{t^{\prime },j}|\leq \left\lceil \frac{m}{t^{\prime }}\right\rceil$ and so, when represented as a binary sequence, the length $|\bar{h}\left(x\right)|$ of $\bar{h}\left(x\right)$ satisfies the following (throughout this paper, for any given q≥2, if needed, we will view a positive integer *m* as a *q*-ary sequence which is the *q*-base representation of *m*, and conversely, we also view a *q*-ary sequence z as a positive integer whose *q*-base representation is z):$$\begin{array}{cc} \left|\bar{h}\left(x\right)\right| & =\sum_{t^{\prime }=1}^{t}\sum_{j=1}^{t^{\prime }}\left|{\bar{h}}_{t^{\prime },j}\left(x\right)\right|\\ \; & \leq \sum_{t^{\prime }=1}^{t}\sum_{j=1}^{t^{\prime }}\left(\log\left\lceil \frac{m}{t^{\prime }}\right\rceil +\log q\right)\\ \; & \leq \frac{t(t+1)}{2}\left(\log m+\log q\right)\\ & =\bar{R}. \end{array}$$Hence, viewed as a positive integer, we have $\bar{h}\left(x\right)\in \left[0,2^{\bar{R}}-1\right]$ for any $x\in \Sigma_{q}^{m}$. Moreover, for each $t^{\prime }\in \left[t\right]$, if $y\in B_{t^{\prime }}\left(x\right)$, then we have $y_{I_{t^{\prime },j}^{\prime }}\in D_{1}\left(x_{I_{t^{\prime },j}}\right)$ for each $j\in [t^{\prime }]$, where $I_{t^{\prime },j}^{\prime }=\left\{\ell \in \left[n-t^{\prime }\right]:\ell \equiv j\;\left(\mathrm{mod}\;t^{\prime }\right)\right\}$. By Lemma 1, $x_{I_{t^{\prime },j}}$ can be recovered from ${\bar{h}}_{t^{\prime },j}\left(x\right)$ and $y_{I_{t^{\prime },j}^{\prime }}$, and so x can be recovered from y and $\bar{h}\left(x\right)$. Equivalently, for any $x^{\prime }\in N_{t}\left(x\right)$, we have $\bar{h}\left(x\right)\neq \bar{h}\left(x^{\prime }\right)$.For each $x\in \Sigma_{q}^{m}$, let P(x) be the set of all positive integers *j* such that *j* is a divisor of $|\bar{h}\left(x\right)-\bar{h}\left(x^{\prime }\right)|$ for some $x^{\prime }\in N_{t}\left(x\right)$. By the same discussions as in the proof of [33] (Lemma 4), we can obtain $\left|P\left(x\right)\right|\leq 2^{\log|N_{t}\left(x\right)|+o\left(\log m\right)}\leq O\left(q^{t}m^{3}\right)$. (Note that *q* and *t* are assumed to fixed integers and, by (Equation 1), $\left|N_{t}(x)|\leq tm^{2}q^{t}\right)$. So, by brute force search, one can find, in time $2^{\log|N_{t}\left(x\right)|+o\left(\log m\right)}\leq O\left(q^{t}m^{3}\right)$, a positive integer $\alpha \left(x\right)\leq 2^{\log|N_{t}\left(x\right)|+o\left(\log m\right)}$ such that α(x)∉P(x). Let $h_{\mathrm{syn}}\left(x\right)=\left(\alpha \left(x\right),\bar{h}\left(x\right)\;\;\mathrm{mod}\;\alpha \left(x\right)\right)$. Then, we have $h_{\mathrm{syn}}\left(x\right)\neq h_{\mathrm{syn}}\left(x^{\prime }\right)$ for all $x^{\prime }\in N_{t}\left(x\right)$. Equivalently, $h_{\mathrm{syn}}\left(x\right)\neq h_{\mathrm{syn}}\left(x^{\prime }\right)$ for any distinct $x,x^{\prime }\in \Sigma_{q}^{n}$ with $B_{\leq t}\left(x\right)\cap B_{\leq t}\left(x^{\prime }\right)\neq Ø$.Finally, since $\alpha \left(x\right)\leq 2^{\log|N_{t}\left(x\right)|+o\left(\log m\right)}$ is a positive integer, and by (Equation 1), $|N_{t}\left(x\right)|\leq tm^{2}q^{t}$, so viewed as a *q*-ary sequence, we have $h_{\mathrm{syn}}\left(x\right)\in \Sigma_{q}^{4\log_{q}m+o\left(\log_{q}m\right)}$.    □

Clearly, for any $x\in \Sigma_{q}^{m}$, given $h_{\mathrm{syn}}\left(x\right)$ and any $y\in B_{\leq t}\left(x\right)$, one can uniquely recover x. This is because for any $x\neq x^{\prime }\in \Sigma_{q}^{m}$ such that $y\in B_{\leq t}\left(x^{\prime }\right)$, we have $y\in B_{\leq t}\left(x\right)\cap B_{\leq t}\left(x^{\prime }\right)$, so $B_{\leq t}\left(x\right)\cap B_{\leq t}\left(x^{\prime }\right)\neq Ø$ and by Lemma 2, we have $h_{\mathrm{syn}}\left(x\right)\neq h_{\mathrm{syn}}\left(x^{\prime }\right)$. Thus, x is uniquely determined by $h_{\mathrm{syn}}\left(x\right)$ and y.

### 2.2. Bounded Burst-Deletion Correction

We give a construction for correcting a single burst-deletion given the knowledge that the location of the deleted symbols are within an interval of length ρ.

Given a positive integer ρ, we define a collection of intervals$$L_{\rho }=\left\{L_{j}:j=1,2,\cdots ,\left\lceil n/\rho \right\rceil -1\right\}$$
such that(2)$$L_{j}=\{\begin{array}{c} \left[\left(j-1\right)\rho +1,\left(j+1\right)\rho \right],\;\;\mathrm{for}\;j\in \left\{1,\cdots ,\left\lceil n/\rho \right\rceil -2\right\},\\ \left[\left(j-1\right)\rho +1,n\right],\;\;\;\;\;\;\;\mathrm{for}\;j=\left\lceil n/\rho \right\rceil -1. \end{array}$$

The following remark is easy to see.

**Remark** **1.**
*The intervals in $L_{\rho }$ satisfy the following:*
*1)* 
*For any interval L⊆[n] of length |L|≤ρ, there is a (not necessarily unique) $j_{0}\in \left[\left\lceil n/\rho \right\rceil -1\right]=\left\{1,2,\cdots ,\left\lceil n/\rho \right\rceil -1\right\}$, such that $L\subseteq L_{j_{0}}$.*
*2)* 
*$L_{j}\cap L_{j^{\prime }}=Ø$ for all $j,j^{\prime }\in \left[1,\left\lceil n/\rho \right\rceil -1\right]$, such that $\left|j-j^{\prime }\right|\geq 2$.*



**Construction 1**: Let $h:\Sigma_{q}^{m}\to \Sigma_{q}^{R_{m}}$ be a function for any positive integer *m*, where $R_{m}$ is a positive integer depending on m,q and *t*, such that $h\left(z\right)\neq h\left(z^{\prime }\right)$ for any distinct $z,z^{\prime }\in \Sigma_{q}^{m}$ with $B_{\leq t}\left(z\right)\cap B_{\leq t}\left(z^{\prime }\right)\neq Ø$. Let $L_{\rho }=\left\{L_{j}:j=1,2,\cdots ,\left\lceil n/\rho \right\rceil -1\right\}$, such that each $L_{j}$ is defined by (Equation 2). For each $x\in \Sigma_{q}^{n}$ and each ℓ∈{0,1}, let(3)$$\begin{array}{c} {\tilde{h}}_{\rho }^{\left(\ell \right)}\left(x\right)=\sum_{\begin{array}{c} j\in [1,\left\lceil n/\rho \right\rceil -1\}:\\ j\;\;\equiv \;\;\ell \;\mathrm{mod}\;2 \end{array}}h\left(x_{L_{j}}\right)\;\left(\mathrm{mod}\;q^{R_{2\rho }}\right). \end{array}$$

The modular operation $\left(\mathrm{mod}\;q^{R_{2\rho }}\right)$ in (Equation 3) is performed on the result of the summation, but not on each $h(x_{L_{j}})$.

**Lemma** **3.**
*Suppose $x\neq x^{\prime }\in \Sigma_{q}^{n}$. Suppose there exists an interval L⊆[n] of length |L|≤ρ and two intervals $D,D^{\prime }\subseteq L$ of size $\left|D|=\right|D^{\prime }|\leq t$, such that $x_{[n]\setminus D}=x_{\left[n\right]\setminus D^{\prime }}^{\prime }$. Then, we have ${\tilde{h}}_{\rho }^{\left(\ell \right)}\left(x\right)\neq {\tilde{h}}_{\rho }^{\left(\ell \right)}\left(x^{\prime }\right)$ for some ℓ∈{0,1}.*


**Proof.** By (Equation 2), for each $j\in \{1,2,\cdots ,\left\lceil n/\rho \right\rceil -1\}$, the length $\left|L_{j}\right|$ of $L_{j}$ satisfies $|L_{j}|\leq 2\rho$. So, for any $x\in \Sigma_{q}^{m}$, the length $\left|h(x_{L_{j}})\right|$ of $h\left(x_{L_{j}}\right)$ (as a *q*-ary sequence) satisfies $|h(x_{L_{j}})|=R_{2|L_{j}|}\leq R_{2\rho }$, which implies that (as an integer),$$h\left(x_{L_{j}}\right)<q^{R_{2\rho }}.$$Since |L|≤ρ, by 1) of Remark 1, there is a $j_{0}\in \left[\left\lceil n/\rho \right\rceil -1\right]=\left\{1,2,\cdots ,\left\lceil n/\rho \right\rceil -1\right\}$, such that $L\subseteq L_{j_{0}}$. Let ℓ∈{0,1} be such that $\ell \equiv j_{0}\;\mathrm{mod}\;2$, and let$$\Lambda_{\ell }=\left\{j\in \left[1,\left\lceil n/\rho \right\rceil -1\right]:j\equiv j_{0}\;\mathrm{mod}\;2\right\}.$$Then, by 2) of Remark 1, $L_{j}\cap L_{j_{0}}=Ø$ for all $j\in \Lambda_{\ell }\setminus \left\{j_{0}\right\}$. Further, by assumption of $D,D^{\prime }$ and *L*, we have $x_{L_{j_{0}}\setminus D}=x_{L_{j_{0}}\setminus D^{\prime }}^{\prime }\in B_{\leq t}\left(x_{L_{j_{0}}}\right)\cap B_{\leq t}\left(x_{L_{j_{0}}}^{\prime }\right)$ and $x_{L_{j}}=x_{L_{j}}^{\prime }$ for all $j\in \Lambda_{\ell }\setminus \left\{j_{0}\right\}$. Therefore, we have$$h\left(x_{L_{j_{0}}}^{\prime }\right)\neq h\left(x_{L_{j_{0}}}\right)$$
and$$h\left(x_{L_{j}}^{\prime }\right)=h\left(x_{L_{j}}\right),\;\;\forall \;j\in \Lambda_{\ell }\setminus \left\{j_{0}\right\}.$$By the above discussions, and by Construction 1, we can obtain that ${\tilde{h}}_{\rho }^{\left(\ell \right)}\left(x\right)\neq {\tilde{h}}_{\rho }^{\left(\ell \right)}\left(x^{\prime }\right)$, which completes the proof.    □

**Remark** **2.**
*Let h be the function $h_{syn}$ constructed in Lemma 2. Then, we have $R_{2\rho }=4\log_{q}\left(2\rho \right)+o\left(\log_{q}\left(2\rho \right)\right)$. Moreover, by Lemma 2, h is computable in time $O(q^{t}{\left(2\rho \right)}^{3})$.*


## 3. Pattern Dense Sequences

The concept of (p,δ)-dense sequences was introduced in [18], and was used to construct binary codes with redundancy $\log n+\frac{t(t+1)}{2}\log\log n+c_{t}$ for correcting a burst of at most *t* deletions, where *n* is the message length and $c_{t}$ is a constant only depending on *t*. In this section, we generalize the (p,δ)-density to *q*-ary sequences and derive some important properties for these sequences that will be used in our new construction in the next section.

The *q*-ary (p,δ)-dense sequences can be defined similarly to the binary (p,δ)-dense sequences as follows.

**Definition** **1.***Let d≤δ≤n be three positive integers and $p\in \Sigma_{q}^{d}$ called a* pattern. *A sequence $x\in \Sigma_{q}^{n}$ is said to be*
(p,δ)-dense *if each substring of x of length δ contains at least one p. The indicator vector of x with respect to p is a vector*$$1_{p}\left(x\right)=\left(1_{p}{\left(x\right)}_{1},1_{p}{\left(x\right)}_{2},\ldots ,1_{p}{\left(x\right)}_{n}\right)\in \Sigma_{2}^{n}$$*such that, for each i∈[n], $1_{p}{\left(x\right)}_{i}=1$ if $x_{\left[i,i+d-1\right]}=p$, and $1_{p}{\left(x\right)}_{i}=0$ otherwise.*

In this section, we will always let (d=2t)$$p=0^{t}1^{t}$$
and view $p=0^{t}1^{t}\in \Sigma_{q}^{2t}$ for any q≥2. Moreover, from Definition 1, we have the following simple remark.

**Remark** **3.**
*Each sequence $x\in \Sigma_{q}^{n}$ can be written as the form $x=x_{0}px_{1}px_{2}p\cdots x_{m-1}px_{m}$ for some m≥0, such that each $x_{i}$, i∈[0,m], is a (possibly empty) string that does not contain p. Moreover, x is (p,δ)-dense if and only if it satisfies the following: (1) the lengths of $x_{0}$ and $x_{m}$ are not greater than δ−2t; and (2) the length of each $x_{i}$, i∈[1,m−1], is not greater than δ+1−4t.*


In [18], the VT syndrome of $a_{p}\left(x\right)$ was used to bound the location of deletions for (p,δ)-dense x, where $a_{p}\left(x\right)$ is a vector of length $n_{p}\left(x\right)+1$, whose *i*-th entry is the distance between positions of the *i*-th and (i+1)-st 1 in the string $\left(1,1_{p}\left(x\right),1\right)$, and $n_{p}\left(x\right)$ is the number of 1s in $1_{p}\left(x\right)$. In this paper, we prove that the VT syndrome of $1_{p}\left(x\right)$ plays the same role. Specifically, for each $x\in \Sigma_{q}^{n}$, let(4)$$\begin{array}{c} a_{0}\left(x\right)=\sum_{i=1}^{n}1_{p}{\left(x\right)}_{i} \end{array}$$
and(5)$$\begin{array}{c} a_{1}\left(x\right)=\sum_{i=1}^{n}i\;\cdot \;1_{p}{\left(x\right)}_{i} \end{array}$$
where $1_{p}\left(x\right)$ is the indicator vector of x with respect to p, as defined in Definition 1. Then, we have the following lemma.

**Lemma** **4.**
*Suppose $x\in \Sigma_{q}^{n}$ is (p,δ)-dense. For any $t^{\prime }\in \left[t\right]$ and any $y\in B_{t^{\prime }}\left(x\right)$, given $a_{0}\left(x\right)\;\left(\mathrm{mod}\;4\right)$ and $a_{1}\left(x\right)\;\left(\mathrm{mod}\;2n\right)$, one can find, in time O(n), an interval L⊆[n] of length |L|≤3δ, such that $y=x_{[n]\setminus D}$ for some interval D⊆L of size $\left|D\right|=t^{\prime }=\left|x\right|-\left|y\right|$ (in fact, we can require that the length of L is at most δ. However, the proof for |L|≤δ needs more careful discussions).*


**Proof.** Let $a_{0}\left(x\right)=m$ and $a_{0}\left(y\right)=m^{\prime }$. Then, by Remark 3, x and y can be written as the following form:$$x=x_{0}0^{t}1^{t}x_{1}0^{t}1^{t}x_{2}\cdots 0^{t}1^{t}x_{m-1}0^{t}1^{t}x_{m}$$
and
$$y=y_{0}0^{t}1^{t}y_{1}0^{t}1^{t}y_{2}\cdots 0^{t}1^{t}y_{m^{\prime }-1}0^{t}1^{t}y_{m^{\prime }}$$
where $x_{i}$ and $y_{j}$ do not contain $p=0^{t}1^{t}$ for each i∈[0,m] and $j\in [0,m^{\prime }]$. We denote$$u_{i}=\left|y_{0}0^{t}1^{t}y_{1}0^{t}1^{t}\cdots y_{i-1}0^{t}1^{t}\right|,\;\forall i\in \left[1,m^{\prime }\right]$$
and
$$v_{i}=\left|y_{0}0^{t}1^{t}y_{1}0^{t}1^{t}\cdots y_{i}\right|,\;\forall i\in \left[0,m^{\prime }\right].$$
Additionally, let $u_{0}=0$. Clearly, for each $i\in [0,m^{\prime }]$, we have $u_{i}\leq v_{i}$ and $y_{i}=y_{\left[u_{i}+1,v_{i}\right]}$. Moreover, for each $i\in [0,m^{\prime }-1]$, each $j_{i}\in \left[u_{i},v_{i}\right]$ and $j_{i+1}\in \left[u_{i+1},v_{i+1}\right]$, we have(6)$$\begin{array}{c} j_{i+1}-j_{i}\geq u_{i+1}-v_{i}\geq 2t. \end{array}$$Note that a burst of $t^{\prime }\leq t$ deletions may destroy at most two ***p***s or create at most one p, so $\Delta_{0}≜m-m^{\prime }\in \left\{-1,0,1,2\right\}$ and $\Delta_{0}$ can be computed from $a_{0}\left(x\right)-a_{0}\left(y\right)$. We need to consider the following four cases according to $\Delta_{0}$.Case 1: $\Delta_{0}=2$. Then, $m^{\prime }=m-2$ and there is an $i_{d}\in \left[0,m^{\prime }\right]$ such that $|x_{i_{d}+1}|\leq t^{\prime }-2$ and y can be obtained from x by deleting a substring $1^{t_{1}}x_{i_{d}+1}0^{t_{0}}$ for some $t_{0},t_{1}>0$, such that $|x_{i_{d}+1}|+t_{0}+t_{1}=t^{\prime }$. More specifically, $y_{i_{d}}=x_{i_{d}}0^{t}1^{t-t_{1}}0^{t-t_{0}}1^{t}x_{i_{d}+2}$. Clearly, we have $2\leq t^{\prime }\leq t$ and $x_{\left[u_{i_{d}}+1,v_{i_{d}}+t^{\prime }\right]}=x_{i_{d}}0^{t}1^{t}x_{i_{d}+1}0^{t}1^{t}x_{i_{d}+2}$. It is sufficient to let $L=[u_{i_{d}}+1,v_{i_{d}}+t^{\prime }]$, but we still need to find $i_{d}$.Consider $1_{p}\left(x\right)$ and $1_{p}\left(y\right)$. By Definition 1, $1_{p}\left(x\right)$ can be obtained from $1_{p}\left(y\right)$ by $t^{\prime }$ insertions and two substitutions in the substring $1_{p}{\left(y\right)}_{\left[u_{i_{d}}+1,v_{i_{d}}\right]}$: inserting $t^{\prime }$ 0s and substituting two 0s by two 1s. Then, by (Equation 5), we can obtain(7)$$\begin{array}{c} a_{1}\left(x\right)=a_{1}\left(y\right)+\lambda_{1}\left(i_{d}\right)+\lambda_{2}\left(i_{d}\right)+\left(m^{\prime }-i_{d}\right)t^{\prime } \end{array}$$
where $\lambda_{1}\left(i_{d}\right),\lambda_{2}\left(i_{d}\right)\in \left[u_{i_{d}}+1,v_{i_{d}}+t^{\prime }\right]$ are the locations of the two substitutions. To find $i_{d}$, we define a function $\xi_{2}$ as follows: for every $i\in [0,m^{\prime }]$, let$$\xi_{2}\left(i\right)=a_{1}\left(y\right)+2\left(u_{i}+1\right)+\left(m^{\prime }-i\right)t^{\prime }.$$Then, for each $i\in [0,m^{\prime }-1]$, we can obtain $\xi_{2}\left(i+1\right)-\xi_{2}\left(i\right)=2\left(u_{i+1}-u_{i}\right)-t^{\prime }\geq 4t-t^{\prime }>0$, where the first inequality comes from (Equation 6). So, for each $i\in [0,m^{\prime }-1]$, we have(8)$$\begin{array}{c} a_{1}\left(y\right)<\xi_{2}\left(i\right)<\xi_{2}\left(i+1\right)\leq \xi_{2}\left(m^{\prime }\right)<a_{1}\left(y\right)+2n, \end{array}$$
where the last inequality comes from the simple observation that $\xi_{2}\left(m^{\prime }\right)=a_{1}\left(y\right)+2\left(u_{m^{\prime }}+1\right)<a_{1}\left(y\right)+2n$.By definition of $\xi_{2}$ and $a_{1}$, we can obtain$$\begin{array}{cc} \xi_{2}\left(i_{d}+1\right)-a_{1}\left(x\right) & =2\left(u_{i_{d}+1}+1\right)-\lambda_{1}\left(i_{d}\right)-\lambda_{2}\left(i_{d}\right)-t^{\prime }\\ \; & \overset{\left(i\right)}{\geq }2u_{i_{d}+1}+2-2\left(v_{i_{d}}+t^{\prime }\right)-t^{\prime }\\ \; & \overset{\left(\mathrm{ii}\right)}{\geq }4t+2-3t^{\prime }\\ & >0 \end{array}$$
where (i) holds because $\lambda_{1}\left(i_{d}\right),\lambda_{2}\left(i_{d}\right)\in \left[u_{i_{d}}+1,v_{i_{d}}+t^{\prime }\right]$, and (ii) is obtained from (Equation 6). On the other hand, by (Equation 7), $a_{1}\left(x\right)-\xi_{2}\left(i_{d}\right)=\lambda_{1}\left(i_{d}\right)+\lambda_{2}\left(i_{d}\right)-2\left(u_{i_{d}}+1\right)\geq 0$ (noticing that $\lambda_{1}\left(i_{d}\right),\lambda_{2}\left(i_{d}\right)\in \left[u_{i_{d}}+1,v_{i_{d}}+t^{\prime }\right])$. Hence, we can obtain(9)$$\begin{array}{c} \xi_{2}\left(i_{d}\right)\leq a_{1}\left(x\right)<\xi_{2}\left(i_{d}+1\right). \end{array}$$By (Equation 8) and (Equation 9), $i_{d}$ and *L* can be found as follows: Compute$$\mu ≜a_{1}\left(x\right)\;\left(\mathrm{mod}\;2n\right)-a_{1}\left(y\right)\;\left(\mathrm{mod}\;2n\right)$$
and$$\mu_{i}≜\xi_{2}\left(i\right)\;\left(\mathrm{mod}\;2n\right)-a_{1}\left(y\right)\;\left(\mathrm{mod}\;2n\right)$$
for *i* from 0 to $m^{\prime }$. Then, we can find an $i_{d}\in \left[0,m^{\prime }\right]$ such that $\mu_{i_{d}}\leq \mu <\mu_{i_{d}+1}$, where $\mu_{m^{\prime }+1}=2n$. Let $L=[u_{i_{d}}+1,v_{i_{d}}+t^{\prime }]$. Note that $x_{\left[u_{i_{d}}+1,v_{i_{d}}+t^{\prime }\right]}=x_{i_{d}}0^{t}1^{t}x_{i_{d}+1}0^{t}1^{t}x_{i_{d}+2}$ and x is (p,δ)-dense, so by Remark 3, the length of *L* satisfies $\left|L|=\right|x_{i_{d}}0^{t}1^{t}x_{i_{d}+1}0^{t}1^{t}x_{i_{d}+2}|\leq 3\left(\delta +1-4t\right)+4t\leq 3\delta$, where the last inequality holds because $2\leq t^{\prime }\leq t$.Case 2: $\Delta_{0}=1$. Then, $m^{\prime }=m-1$ and, similarly to Case 1, there is an $i_{d}\in \left[0,m^{\prime }\right]$ such that $y_{i_{d}}$ can be obtained from $x_{i_{d}}0^{t}1^{t}x_{i_{d}+1}$ by deleting $t^{\prime }$ symbols and the pattern $0^{t}1^{t}$ is destroyed. Clearly, $x_{\left[u_{i_{d}}+1,v_{i_{d}}+t^{\prime }\right]}=x_{i_{d}}0^{t}1^{t}x_{i_{d}+1}$, and it is sufficient to let $L=[u_{i_{d}}+1,v_{i_{d}}+t^{\prime }]$. To find $i_{d}$, consider $1_{p}\left(y\right)$ and $1_{p}\left(x\right)$. By Definition 1, $1_{p}\left(x\right)$ can be obtained from $1_{p}\left(y\right)$ by $t^{\prime }$ insertions and one substitution in the substring $1_{p}{\left(y\right)}_{\left[u_{i_{d}}+1,v_{i_{d}}\right]}$: inserting $t^{\prime }$ 0s and substituting a 0 by a 1. By (Equation 5), we can obtain(10)$$\begin{array}{c} a_{1}\left(x\right)=a_{1}\left(y\right)+\lambda \left(i_{d}\right)+\left(m^{\prime }-i_{d}\right)t^{\prime } \end{array}$$
where $\lambda \left(i_{d}\right)\in \left[u_{i_{d}}+1,v_{i_{d}}+t^{\prime }\right]$ is the location of the substitution. For every $i\in [0,m^{\prime }]$, let$$\xi_{1}\left(i\right)=a_{1}\left(y\right)+\left(u_{i}+1\right)+\left(m^{\prime }-i\right)t^{\prime }.$$Then, for each $i\in [0,m^{\prime }-1]$, we have $\xi_{1}\left(i+1\right)-\xi_{1}\left(i\right)=u_{i+1}-u_{i}-t^{\prime }\geq 2t-t^{\prime }>0$, and so we can further obtain(11)$$\begin{array}{c} a_{1}\left(y\right)<\xi_{1}\left(i\right)<\xi_{1}\left(i+1\right)\leq \xi_{1}\left(m^{\prime }\right)\leq a_{1}\left(y\right)+n. \end{array}$$By definition of $\xi_{1}$ and $a_{1}$, we can obtain $\xi_{1}\left(i_{d}+1\right)-a_{1}\left(x\right)=u_{i_{d}+1}+1-\lambda \left(i_{d}\right)-t^{\prime }>u_{i_{d}+1}+1-\left(v_{i}+t^{\prime }\right)-t^{\prime }\geq 2t+1-2t^{\prime }>0$. On the other hand, by (Equation 10), $a_{1}\left(x\right)-\xi_{1}\left(i_{d}\right)=\lambda \left(i_{d}\right)-\left(u_{i_{d}}+1\right)\geq 0$. Hence, we can obtain(12)$$\begin{array}{c} \xi_{1}\left(i_{d}\right)\leq a_{1}\left(x\right)<\xi_{1}\left(i_{d}+1\right). \end{array}$$By (Equation 11) and (Equation 12), *L* can be found as follows: Compute$$\mu ≜a_{1}\left(x\right)\;\left(\mathrm{mod}\;2n\right)-a_{1}\left(y\right)\;\left(\mathrm{mod}\;2n\right)$$
and$$\mu_{i}≜\xi_{1}\left(i\right)\;\left(\mathrm{mod}\;2n\right)-a_{1}\left(y\right)\;\left(\mathrm{mod}\;2n\right)$$
for *i* from 0 to $m^{\prime }$. Let $i_{d}\in \left[0,m^{\prime }\right]$ be such that $\mu_{i_{d}}\leq \mu <\mu_{i_{d}+1}$. Then, let $L=[u_{i_{d}}+1,v_{i_{d}}+t^{\prime }]$, where $\mu_{m^{\prime }+1}=2n$. Note that $x_{\left[u_{i_{d}}+1,v_{i_{d}}+t^{\prime }\right]}=x_{i_{d}}0^{t}1^{t}x_{i_{d}+1}$ and x is (p,δ)-dense, so by Remark 3, $\left|L|=\right|x_{i_{d}}0^{t}1^{t}x_{i_{d}+1}|\leq 2\left(\delta +1-4t\right)+2t<2\delta$.Case 3: $\Delta_{0}=0$. Then, $m^{\prime }=m$. For every i∈[0,m], let$$\xi_{0}\left(i\right)=a_{1}\left(y\right)+\left(m-i\right)t^{\prime }.$$Note that x contains *m* copies of $0^{t}1^{t}$, so we have $n\geq 2tm>mt^{\prime }$. Therefore, for each i∈[0,m−1], we can obtain(13)$$\begin{array}{c} a_{1}\left(y\right)+n>a_{1}\left(y\right)+mt^{\prime }\geq \xi_{0}\left(i\right)>\xi_{0}\left(i+1\right)\geq a_{1}\left(y\right). \end{array}$$As $\Delta_{0}=0$, there are two ways to obtain y from x:
1)There is an $i_{d}\in \left[0,m\right]$ such that $y_{i_{d}}$ can be obtained from $x_{i_{d}}$ by a burst of $t^{\prime }$ deletions. Correspondingly, by Definition 1, $1_{p}\left(x\right)$ can be obtained from $1_{p}\left(y\right)$ by inserting $t^{\prime }$ 0s into $1_{p}{\left(y\right)}_{\left[u_{i_{d}}+1,v_{i_{d}}\right]}$. Therefore, we have(14)$$\begin{array}{c} a_{1}\left(x\right)=a_{1}\left(y\right)+\left(m-i_{d}\right)t^{\prime }=\xi_{0}\left(i_{d}\right). \end{array}$$2)There is an $i_{d}\in \left[0,m-1\right]$ such that $x_{i_{d}}0^{t}1^{t}x_{i_{d}+1}=y_{i_{d}}0^{t+t_{0}}1^{t+t_{1}}y_{i_{d}+1}$ for some $t_{0},t_{1}\in \left[1,t^{\prime }-1\right]$, such that $t_{0}+t_{1}=t^{\prime }$, and $y_{i_{d}}0^{t}1^{t}y_{i_{d}+1}$ is obtained from $x_{i_{d}}0^{t}1^{t}x_{i_{d}+1}$ by deleting the substring $0^{t_{0}}1^{t_{1}}$. By Definition 1, $1_{p}\left(x\right)$ can be obtained from $1_{p}\left(y\right)$ by inserting $t_{0}$ 0s in $1_{p}{\left(y\right)}_{\left[u_{i_{d}}+1,v_{i_{d}}\right]}$ and $t_{1}$ 0s in $1_{p}{\left(y\right)}_{\left[v_{i_{d}}+2,v_{i_{d}+2t}\right]}$. Therefore, we have$$a_{1}\left(x\right)=a_{1}\left(y\right)+t_{0}+\left(m-i_{d}-1\right)t^{\prime }.$$By definition of $\xi_{0}$, we have $\xi_{0}\left(i_{d}\right)-a_{1}\left(x\right)=t^{\prime }-t_{0}>0$ and $a_{1}\left(x\right)-\xi_{0}\left(i_{d}+1\right)=t_{0}>0$. So, we can obtain(15)$$\begin{array}{c} \xi_{0}\left(i_{d}\right)>a_{1}\left(x\right)>\xi_{0}\left(i_{d}+1\right) \end{array}$$For both cases, if $i_{d}\in \left[0,m-1\right]$, then we can $L=[u_{i_{d}}+1,v_{i_{d}}+2t+t^{\prime }]$; if $i_{d}=m$, then we can let $L=[u_{m}+1,n]$. Note that $x_{\left[u_{i_{d}}+1,v_{i_{d}}+2t+t^{\prime }\right]}=x_{i_{d}}0^{t}1^{t}$ and $x_{\left[u_{m}+1,n\right]}=x_{m}$, and since x is (p,δ)-dense, then by Remark 3, we have $\left|L|=\right|x_{i_{d}}0^{t}1^{t}|\leq 2\delta$ or $\left|L|=\right|x_{m}|\leq 2\delta$. Moreover, by (Equation 13), (Equation 14), and (Equation 15), $i_{d}$ (and so L) can be found as follows: Compute$$\mu ≜a_{1}\left(x\right)\;\left(\mathrm{mod}\;2n\right)-a_{1}\left(y\right)\;\left(\mathrm{mod}\;2n\right)$$
and$$\mu_{i}≜\xi_{0}\left(i\right)\;\left(\mathrm{mod}\;2n\right)-a_{1}\left(y\right)\;\left(\mathrm{mod}\;2n\right)$$
for *i* from 0 to *m*. Then, we can always find an $i_{d}\in \left[0,m\right]$, such that $\mu_{i_{d}}\geq \mu >\mu_{i_{d}+1}$, which is what we want.Case 4: $\Delta_{0}=-1$. Then, $m^{\prime }=m+1$ and there is an $i_{d}\in \left[0,m^{\prime }-1\right]$ such that $x_{i_{d}}=y_{i_{d}}0^{t_{0}}s0^{t-t_{0}}1^{t}y_{i_{d}+1}$ or $x_{i_{d}}=y_{i_{d}}0^{t}1^{t_{1}}s1^{t-t_{1}}y_{i_{d}+1}$, where $t_{0}\in \left[1,t\right]$, $t_{1}\in \left[1,t-1\right]$ and $s\in \Sigma_{q}^{t^{\prime }}$, and y can be obtained from x by deleting s. In this case, we can let $L=[v_{i_{d}}+1,v_{i_{d}}+2t+t^{\prime }]$, and can obtain $\left|L\right|=2t+t^{\prime }<\delta$. To find $i_{d}$, we consider $1_{p}\left(x\right)$ and $1_{p}\left(y\right)$. By Definition 1, $1_{p}\left(x\right)$ can be obtained from $1_{p}\left(y\right)$ by inserting $t^{\prime }$ 0s into $1_{p}{\left(y\right)}_{\left[v_{i_{d}+1},v_{i_{d}}+2t\right]}$ and substituting $1_{p}{\left(y\right)}_{v_{i_{d}}+1}=1$ by a 0. Therefore, we have(16)$$\begin{array}{c} a_{1}\left(x\right)=a_{1}\left(y\right)-\left(v_{i_{d}}+1\right)+\left(m^{\prime }-1-i_{d}\right)t^{\prime }. \end{array}$$For every $i\in [0,m^{\prime }-1]$, let$$\xi_{-1}\left(i\right)=a_{1}\left(y\right)-\left(v_{i}+1\right)+\left(m^{\prime }-1-i\right)t^{\prime }.$$Then, for each $i\in [0,m^{\prime }-2]$, we have $\xi_{-1}\left(i\right)-\xi_{-1}\left(i+1\right)=v_{i+1}-v_{i}-t^{\prime }>0$, where the inequality is obtained from (Equation 6). Moreover, we have $\xi_{-1}\left(0\right)=a_{1}\left(y\right)-1+\left(m^{\prime }-1\right)t^{\prime }<a_{1}\left(y\right)+2tm^{\prime }<a_{1}\left(y\right)+n$ and $\xi_{-1}\left(m^{\prime }-1\right)=a_{1}\left(y\right)-\left(v_{m^{\prime }-1}+1\right)>a_{1}\left(y\right)-n$. So, for each $i\in [0,m^{\prime }-2]$, we can obtain(17)$$\begin{array}{c} a_{1}\left(y\right)+n>\xi_{-1}\left(i\right)>\xi_{-1}\left(i+1\right)>a_{1}\left(y\right)-n. \end{array}$$By (Equation 16), and by the definition of $\xi_{-1}$, we have $a_{1}\left(x\right)=\xi_{-1}\left(i_{d}\right)$. So, by (Equation 17), $i_{d}$ (and so L) can be found by the following process: For *i* from 0 to $m^{\prime }-1$, compute $\xi_{-1}\left(i\right)$. Then, we can always find an $i_{d}\in \left[0,m^{\prime }-1\right]$ such that $\xi_{-1}\left(i_{d}\right)\;\left(\mathrm{mod}\;2n\right)=a_{1}\left(x\right)\;\left(\mathrm{mod}\;2n\right)$, which is what we want.Thus, one can always find the expected interval L⊆[n]. From the above discussions, it is easy to see that the time complexity for finding such *L* is O(n).    □

In the rest of this section, we will use the so-called sequence replacement (SR) technique to construct *q*-ary (p,δ)-dense strings with only one symbol of redundancy for $\delta =2tq^{2t}\left\lceil \log n\right\rceil$. The SR technique, which has been widely used in the literature (e.g., see [19,34,35,36]), is an efficient method for constructing strings with or without some constraints on their substrings. In this paper, to apply the SR technique to construct (p,δ)-dense strings, each length-δ string that does not contain p needs to be compressed to a shorter sequence, which can be realized by the following lemma.

**Lemma** **5.**
*Let $\delta =2tq^{2t}\left\lceil \log n\right\rceil$ and $S\subseteq \Sigma_{q}^{\delta }$ be the set of all sequences of length δ that do not contain $p=0^{t}1^{t}$. For $n\geq q^{\frac{6t+3-\log_{q}e}{0.4}}$, there exists an invertible function*

$$g:S\to \Sigma_{q}^{\delta -\lceil \log_{q}n\rceil -6t-2}$$

*such that g and $g^{-1}$ are computable in time O(δ).*


**Proof.** The proof follows the same idea of Proposition 1 of [19], and is replicated here for completeness.As each s∈S has length $\delta =2tq^{2t}\left\lceil \log n\right\rceil$ and does not contain p, then S can be viewed as a subset of ${\left(\Sigma_{q}^{2t}\setminus \left\{p\right\}\right)}^{q^{2t}\left\lceil \log n\right\rceil }$, and we have$$\begin{array}{cc} \log_{q}\left|S\right| & \leq \log_{q}{\left(q^{2t}-1\right)}^{q^{2t}\left\lceil \log n\right\rceil }\\ \; & =\left(2t\right)q^{2t}\left\lceil \log n\right\rceil +\left\lceil \log n\right\rceil \log_{q}{\left(1-\frac{1}{q^{2t}}\right)}^{q^{2t}}\\ \; & \overset{\left(i\right)}{\leq }\left(2t\right)q^{2t}\left\lceil \log n\right\rceil +\left(\log n+1\right)\log_{q}\left(\frac{1}{e}\right)\\ \; & =\delta -\log_{q}n\log e-\log_{q}e\\ \; & \leq \delta -1.4\log_{q}n-\log_{q}e\\ \; & \overset{\left(\mathrm{ii}\right)}{\leq }\delta -\left\lceil \log_{q}n\right\rceil -6t-2, \end{array}$$
where (i) comes from the fact that ${\left(1-\frac{1}{x}\right)}^{x}<\frac{1}{e}$ for x≥1, and (ii) holds when $0.4\log_{q}n+\log_{q}e\geq 6t+3$, i.e., $n\geq q^{\frac{6t+3-\log_{q}e}{0.4}}$. Thus, each sequence in S can be represented by a *q*-ary sequence of length $\delta -\lceil \log_{q}n\rceil -6t-2$, which gives an invertible function $g:S\to \Sigma_{q}^{\delta -\lceil \log_{q}n\rceil -6t-2}$.The computation of *g* and $g^{-1}$ involve conversion of integers in $\left[0,{\left(q^{2t}-1\right)}^{q^{2t}\left\lceil \log n\right\rceil }-1\right]$ between $\left(q^{2t}-1\right)$-base representation and *q*-base representation, so have time complexity $O(2tq^{2t}\left\lceil \log n\right\rceil )=O\left(\delta \right)$.    □

In the rest of this section, we will always let$$\delta =2tq^{2t}\left\lceil \log n\right\rceil .$$

As we are interested in large *n*, we will always assume that $n\geq q^{\frac{6t+3-\log_{q}e}{0.4}}$. The following lemma gives a function for encoding *q*-ary strings to (p,δ)-dense strings.

**Lemma** **6.**
*There exists an invertible function, denoted by $\mathrm{EncDen}:\Sigma_{q}^{n-1}\to \Sigma_{q}^{n}$, such that for every $u\in \Sigma_{q}^{n-1}$, x=EncDen(u) is (p,δ)-dense. Both EncDen and its inverse, denoted by DecDen, are computable in O(nlogn) time.*


**Proof.** Let *g* be the function constructed in Lemma 5. The functions EncDen and DecDen are described by Algorithms 1 and 2, respectively, where each integer i∈[n] is also viewed as a *q*-ary string of length ⌈logn⌉ which is the *q*-base representation of *i*.The correctness of Algorithm 1 can be proved as follows:
1)In the initialization step, if $\tilde{u}=u_{\left[n-\delta +2t,n-1\right]}$ contains p then, clearly, x has length *n*. If $\tilde{u}=u_{\left[n-\delta +2t,n-1\right]}$ does not contain p, then$$x=(u_{\left[1,n^{\prime }\right]},p,p,g\left(\left(\tilde{u},0^{2t}\right)\right),0^{\lceil \log_{q}n\rceil +3})$$
and so $\left|x\right|=n^{\prime }+4t+\left|g\left(\left(\tilde{u},0^{2t}\right)\right)\right|+\left\lceil \log_{q}n\right\rceil +3=n$, where $n^{\prime }=n-\delta +2t-1$, and by Lemma 5, $|g\left(\left(\tilde{u},0^{2t}\right)\right)|=\delta -\left\lceil \log_{q}n\right\rceil -6t-2$. So, at the end of the initialization step, x has length *n*. Moreover, $x_{\left[n^{\prime }+1,n^{\prime }+2t\right]}=p$, and the substring $x_{\left[n^{\prime }+2t+1,n\right]}$ has length ≤δ−4t+1.2)In each round of the replacement step, if $\tilde{x}≜x_{\left[i,i+\delta -1\right]}$ does not contain p for some $i\in [1,n^{\prime }-\delta +1]$, then by Lemma 5, $|\left(p,p,i,g\left(\tilde{x}\right),0,1^{2t},0\right)\left|=\delta =\right|x_{\left[i,i+\delta -1\right]}|$, so by replacement, the length of the appended string equals to the length of the deleted substring, and hence the length of x keeps unchanged.3)At the beginning of each round of the replacement step, we have $x_{\left[n^{\prime }+1,n^{\prime }+2t\right]}=p$, so for $i\in [n^{\prime }+2t-\delta +1,n^{\prime }]$, the substring $x_{\left[i,i+\delta -1\right]}$ contains p. Equivalently, if $\tilde{x}≜x_{\left[i,i+\delta -1\right]}$ does not contain p for some $i\in [n^{\prime }-\delta +2,n^{\prime }]$, then it must be that $i\in [n^{\prime }-\delta +2,n^{\prime }+2t-\delta ]$. In this case, $|\left(p,p,i,g\left(\left(x_{\left[i,n^{\prime }\right]},0^{\ell }\right)\right),0,1^{2t-\ell },0\right)\left|=\delta -\ell =\right|x_{\left[i,n^{\prime }\right]}|$, so by replacement, the length of the appended string equals to the length of the deleted substring, and hence the length of x keeps unchanged.4)By 1), 2) and 3), the substring $x_{\left[n^{\prime }+1,n-\delta +1\right]}$ is always of the form puppv⋯ppw, where all substrings u,v,⋯,w have a length not greater than δ+1−4t. So, by Remark 3, for each $i\in [n^{\prime }+1,n-\delta +1]$, the substring $x_{\left[i,i+\delta -1\right]}$ contains p.5)At the end of each round of the replacement step, the value of $n^{\prime }$ strictly decreases, so the **While** loop will end after at most *n* rounds, and at this time, for each $i\in [1,n^{\prime }]$, the substring $x_{\left[i,i+\delta -1\right]}$ contains p, which combining with 4) implies that x is (p,δ)-dense.The correctness of Algorithm 2 can be easily seen from Algorithm 1, so DecDen is the inverse of EncDen.Note that Algorithms 1 and 2 have at most *n* rounds of replacement, and in each round, g (resp. $g^{-1})$ needs to be computed, which has time complexity O(δ)=O(logn) by Lemma 5, so the total time complexity of Algorithms 1 and 2 is O(nlogn).    □

**Algorithm 1**: The function EncDen for encoding to (p,δ)-dense sequence
1: **Input:**   $u\in \Sigma_{q}^{n-1}$2: **Output:** $x=\mathrm{EncDen}\left(u\right)\in \Sigma_{q}^{n}$ such that x is (p,δ)-dense3: (**Initialization Step:**)4: Let $\tilde{u}=u_{\left[n-\delta +2t,n-1\right]}$.5: **if** $\tilde{u}$ contains p **then**6:      let $n^{\prime }$ be the smallest i∈[n−δ+2t−1,n−2] such that $u_{\left[i+1,i+2t\right]}=p$, and let x=(u,1);7: 
**else**
8:      let $n^{\prime }=n-\delta +2t-1$ and $x=(u_{\left[1,n^{\prime }\right]},p,p,g\left(\left(\tilde{u},0^{2t}\right)\right),0^{\lceil \log_{q}n\rceil +3})$.9: 
**end if**
10: (**Replacement Step:**)11: **while** there exists an $i\in [1,n^{\prime }]$ such that $\tilde{x}≜x_{\left[i,i+\delta -1\right]}$ does not contain p **do**12:     **if** $i\in [1,n^{\prime }-\delta +1]$ **then**13:          delete $x_{\left[i,i+\delta -1\right]}$ from x and append $\left(p,p,i,g\left(\tilde{x}\right),0,1^{2t},0\right)$ to x; let $n^{\prime }=n^{\prime }-\delta$.14:     **end if**15:     **if** $i\in [n^{\prime }-\delta +2,n^{\prime }]$ **then**16:          delete $x_{\left[i,n^{\prime }\right]}$ from x and append $\left(p,p,i,g\left(\left(x_{\left[i,n^{\prime }\right]},0^{\ell }\right)\right),0,1^{2t-\ell },0\right)$ to x, where $\ell ≜\delta -|x_{\left[i,n^{\prime }\right]}|$ satisfying 1≤ℓ≤2t−1; let $n^{\prime }=i-1$.17:     **end if**18: 
**end while**
19: **Return**  x=EncDen(u).


**Algorithm 2**: The function DecDen for decoding of (p,δ)-dense sequence
1: **Input:**   $x=\mathrm{EncDen}\left(u\right)\in \Sigma_{q}^{n}$2: **Output:** 
$u\in \Sigma_{q}^{n-1}$3: **while** $x_{\left[n-\ell^{\prime }-2,n\right]}=01^{\ell^{\prime }}0$ for some $\ell^{\prime }\in \left[1,2t\right]$ **do**4:      let $\tilde{u}$ be obtained from $g^{-1}\left(x_{\left[n-\delta +6t-\ell^{\prime }+1+\left\lceil \log_{q}n\right\rceil ,n-\ell^{\prime }-2\right]}\right)$ by deleting the last $2t-\ell^{\prime }$ symbols; delete the last $\delta +\ell^{\prime }-2t$ symbols of x and insert $\tilde{u}$ at the position *i* of x such that $i=x_{\left[n-\delta +6t-\ell^{\prime }+1,n-\delta +6t-\ell^{\prime }+\left\lceil \log n\right\rceil \right]}$.5: 
**end while**
6: **if** 
$x_{n}=x_{n-1}=0$ 
**then**7:      let $\tilde{u}$ be obtained from $g^{-1}\left(x_{\left[n-\delta +6t,n-\left\lceil \log_{q}n\right\rceil -3\right]}\right)$ by deleting the last 2t 0s and let $u=(x_{\left[1,n-\delta +2t-1\right]},\tilde{u})$.8: 
**end if**
9: **if** 
$x_{n}=1$ 
**then**10:     let $u=x_{\left[1,n-1\right]}$.11: 
**end if**
12: **Return**  u=DecDen(x).


The Algorithm 1 is obtained by modifying the Algorithm 2 of [19], which is for binary sequences but for our purpose we need apply it to *q*-ary sequences.

## 4. Burst-Deletion Correcting q-ary Codes

In this section, we still let$$\delta =2tq^{2t}\left\lceil \log n\right\rceil .$$

Based on (p,δ)-dense sequences, for any fixed positive integers *t* and q≥2, we will construct a family of *q*-ary codes that can correct a burst of at most *t* deletions. The basic idea of our construction is as follows: For each (p,δ)-dense sequence x, use the sketches $a_{0}\left(x\right)$ and $a_{1}\left(x\right)$ (defined by (Equation 4) and (Equation 5), respectively) to locate the deletions within an interval of length 3δ. Then, use the functions ${\tilde{h}}_{\rho }^{\left(0\right)}\left(x\right),{\tilde{h}}_{\rho }^{\left(1\right)}\left(x\right)$ constructed in Construction 1 to uniquely recover x.

Let $h:\Sigma_{q}^{m}\to \Sigma_{q}^{R_{m}}$ be a sketch function for any positive integer *m*, where $R_{m}$ is a positive integer depending on m,q and *t*, such that $h\left(z\right)\neq h\left(z^{\prime }\right)$ for any distinct $z,z^{\prime }\in \Sigma_{q}^{m}$ with $B_{\leq t}\left(z\right)\cap B_{\leq t}\left(z^{\prime }\right)\neq Ø$. For each $x\in \Sigma_{q}^{n}$, let(18)$$\begin{array}{c} f\left(x\right)=\left(a_{0}\left(x\right)\left(\mathrm{mod}\;4\right),a_{1}\left(x\right)\left(\mathrm{mod}\;2n\right),{\tilde{h}}_{\rho }^{\left(0\right)}\left(x\right),{\tilde{h}}_{\rho }^{\left(1\right)}\left(x\right)\right) \end{array}$$
such that $a_{0}\left(x\right)$ is defined by (Equation 4), $a_{1}\left(x\right)$ is defined by (Equation 5), ${\tilde{h}}_{\rho }^{\left(0\right)}\left(x\right)$ and ${\tilde{h}}_{\rho }^{\left(1\right)}\left(x\right)$ are obtained by Construction 1 with $\rho =3\delta =6tq^{2t}\left\lceil \log n\right\rceil$.

**Lemma** **7.**
*Let f be the function given by (Equation 18). If $x\in \Sigma_{q}^{n}$ is (p,δ)-dense, then given f(x) and any $y\in B_{\leq t}\left(x\right)$, one can uniquely recover x.*


**Proof.** First, since x is (p,δ)-dense, then by Lemma 4, given $a_{0}\left(x\right)\;\left(\mathrm{mod}\;4\right)$ and $a_{1}\left(x\right)\;\left(\mathrm{mod}\;2n\right)$, one can find an interval *L* of length |L|≤3δ=ρ, such that $y=x_{[n]\setminus D}$ for some interval D⊆L of size $t^{\prime }=n-\left|y\right|$. Moreover, suppose $x^{\prime }\in \Sigma_{q}^{n}$, such that $y=x_{\left[n\right]\setminus D^{\prime }}^{\prime }$ for some interval $D^{\prime }\subseteq L$ of size $t^{\prime }=n-\left|y\right|$. Then, by Lemma 3, we have ${\tilde{h}}_{\rho }^{\left(\ell \right)}\left(x\right)\neq {\tilde{h}}_{\rho }^{\left(\ell \right)}\left(x^{\prime }\right)$ for some ℓ∈{0,1}. Thus, given $f\left(x\right)=(a_{0}\left(x\right)\;\left(\mathrm{mod}\;4\right),a_{1}\left(x\right)\;\left(\mathrm{mod}\;2n\right),{\tilde{h}}_{\rho }^{\left(0\right)}\left(x\right),{\tilde{h}}_{\rho }^{\left(1\right)}\left(x\right))$ and any $y\in B_{\leq t}\left(x\right)$, one can uniquely recover x. □

Using the function *f*, we can give an encoding function of a *q*-ary code that can correct a burst of at most *t* deletions.

**Lemma** **8.**
*Let f be the function given by (Equation 18) and $f_{q}\left(x\right)$ be the q-ary representation of f(x). Let*

(19)
$$\begin{array}{cc} E:\Sigma_{q}^{n-1} & \to \;\;\;\;\;\Sigma_{q}^{n+r}\\ u\;\; & \mapsto \left(x,0^{t}1,f_{q}\left(x\right)\right) \end{array}$$

*such that x=EncDen(u) and $r=t+1+|f_{q}\left(x\right)|$. Then, for each z=E(u), given any $y\in B_{\leq t}\left(z\right)$, one can uniquely recover x (and so z).*


**Proof.** Let $t^{\prime }=\left|z\right|-\left|y\right|$. Suppose $D=\left[i_{d},i_{d}+t^{\prime }-1\right]\subseteq \left[1,n+r\right]$ is an interval such that $y=z_{[n+r]\setminus D}$. If there is more than one interval *D* such that $y=z_{[n+r]\setminus D}$, then we consider the *D* with the smallest $i_{d}$. Clearly, we have $i_{d}\in \left[1,n+r-t^{\prime }+1\right]$. If $i_{d}\in \left[1,n+t+1-t^{\prime }\right]$, then $y_{n+t+1-t^{\prime }}=z_{n+t+1}=1$; if $i_{d}\in \left[n+t+2-t^{\prime },n+r-t^{\prime }+1\right]$, then $y_{n+t+1-t^{\prime }}=z_{n+t+1-t^{\prime }}=0$. So, we have the following two cases.Case 1: $y_{n+t+1-t^{\prime }}=1$. Then, $i_{d}\in \left[1,n+t-t^{\prime }+1\right]$. We need further to consider the following three subcases.Case 1.1: $y_{\left[n+1-t^{\prime },n+1+t-t^{\prime }\right]}=0^{t}1$. In this case, it must be that D⊆[1,n]. Therefore, we have $y_{\left[1,n-t^{\prime }\right]}\in B_{t^{\prime }}\left(x\right)$ and $y_{\left[n+t+2-t^{\prime },n+r-t^{\prime }\right]}=f_{q}\left(x\right)$. By Lemma 7, x can be recovered from $y_{\left[1,n-t^{\prime }\right]}$ and $y_{\left[n+t+2-t^{\prime },n+r-t^{\prime }\right]}$ correctly.Case 1.2: There is a $t^{\prime \prime }\in \left[1,t^{\prime }-1\right]$ such that $y_{\left[n+1-t^{\prime }+t^{\prime \prime },n+1+t-t^{\prime }\right]}=0^{t-t^{\prime \prime }}1$ and $y_{n-t+t^{\prime \prime }}\neq 0$. In this case, it must be that $D=[n+1-t^{\prime }+t^{\prime \prime },n+t^{\prime \prime }]$. Therefore, $y_{\left[1,n+1-t^{\prime }+t^{\prime \prime }\right]}=x_{\left[1,n+1-t^{\prime }+t^{\prime \prime }\right]}\in B_{t^{\prime }-t^{\prime \prime }}\left(x\right)$ and $y_{\left[n+t+2-t^{\prime },n+r-t^{\prime }\right]}=f_{q}\left(x\right)$. By Lemma 7, x can be recovered from $y_{\left[1,n+1-t^{\prime }+t^{\prime \prime }\right]}$ and $y_{\left[n+t+2-t^{\prime },n+r-t^{\prime }\right]}$ correctly.Case 1.3: $y_{\left[n+1,n+1+t-t^{\prime }\right]}=0^{t-t^{\prime }}1$ and $y_{n}\neq 0$. In this case, it must be that D⊆[n+1,n+t]. Therefore, $y_{\left[1,n\right]}=x$.Case 2: $y_{n+t+1-t^{\prime }}=0$. Then, we have $i_{d}\in \left[n+t+2-t^{\prime },n+r-t^{\prime }+1\right]$ and $x=y_{\left[1,n\right]}$.Thus, one can always uniquely recover x from y. □

**Theorem** **1.**
*There exists a q-ary code of length n capable of correcting a burst of at most t deletions, which has redundancy logn+8loglogn+o(loglogn) bits, encoding complexity $O(q^{7t}n{\left(\log n\right)}^{3})$ and decoding complexity O(nlogn).*


**Proof.** In Construction 1, let *h* be the function $h_{\mathrm{syn}}$ given by Lemma 2 and let $C_{\mathrm{syn}}$ be the code with the encoding function E constructed in Lemma 8. Then, $C_{\mathrm{syn}}\subseteq \Sigma_{q}^{n}$ is a code capable of correcting a burst of at most *t* deletions.The redundancy of $C_{\mathrm{syn}}$ is $1+r=1+t+1+|f_{q}\left(x\right)|=t+2+|f_{q}\left(x\right)|$ in *q*-ary symbols. We now evaluate $|f_{q}\left(x\right)|$. By Lemma 2, we have $R_{2\rho }=4\log_{q}\left(2\rho \right)+o\left(\log_{q}\left(2\rho \right)\right)$. Moreover, since $\rho =6tq^{2t}\left\lceil \log n\right\rceil$, then for ℓ∈{0,1}, by (Equation 3) and Remark 2, the length $\left|{\tilde{h}}_{\rho }^{\left(\ell \right)}\left(x\right)\right|$ of ${\tilde{h}}_{\rho }^{\left(\ell \right)}\left(x\right)$ (as a *q*-ary string) satisfies$$\begin{array}{cc} |{\tilde{h}}_{\rho }^{\left(\ell \right)}\left(x\right)| & <R_{2\rho }\\ & =4\log_{q}\left(12tq^{2t}\left\lceil \log n\right\rceil \right)+o\left(\log_{q}\left(12tq^{2t}\left\lceil \log n\right\rceil \right)\right)\\ & =4\log_{q}\log n+o\left(\log_{q}\log n\right). \end{array}$$So, by (Equation 4) and (Equation 5), the length $|f_{q}\left(x\right)|$ of $f_{q}\left(x\right)$ (as a *q*-ary string) satisfies$$\begin{array}{cc} |f_{q}\left(x\right)| & =\log_{q}4+\log_{q}\left(2n\right)+|{\tilde{h}}_{\rho }^{\left(0\right)}\left(x\right)|+|{\tilde{h}}_{\rho }^{\left(1\right)}\left(x\right)|\\ & \leq \log_{q}n+8\log_{q}\log n+o\left(\log_{q}\log n\right). \end{array}$$Thus, the redundancy of $C_{\mathrm{syn}}$, measured in bits, is $(t+2+|f_{q}\left(x\right)|)\log q\leq \log n+8\log\log n+o\left(\log\log n\right)$.Consider the encoding complexity of $C_{\mathrm{syn}}$. Note that, by (Equation 3) and Remark 2, ${\tilde{h}}_{\rho }^{\left(0\right)}\left(x\right)$ and ${\tilde{h}}_{\rho }^{\left(1\right)}\left(x\right)$ are computable in time $O\left(nq^{t}{\left(2\rho \right)}^{3}\right)=O\left(nq^{t}{\left(12tq^{2t}\left\lceil \log n\right\rceil \right)}^{3}\right)=O\left(q^{7t}n{\left(\log n\right)}^{3}\right)$. Then, by (Equation 4), (Equation 5), and (Equation 18), f(x) is computable in time $O(q^{7t}n{\left(\log n\right)}^{3})$. Moreover, by Lemma 6, the mapping EncDen is computable in O(nlogn) time, so by (Equation 19), the encoding complexity of $C_{\mathrm{syn}}$ is $O(q^{7t}n{\left(\log n\right)}^{3})$.For the decoding complexity of $C_{\mathrm{syn}}$, by Lemma 6, the inverse of EncDen is computable in O(nlogn) time, so by the proof of Lemma 8, we only need to consider the complexity of recovering x from f(x) and any given $y\in B_{\leq t}\left(x\right)$. By Lemma 4 and 1) of Remark 1, one can locate the deletions within an interval $L_{j_{0}}$ in time O(n) using $a_{0}\left(x\right)\left(\mathrm{mod}\;4\right)$ and $a_{1}\left(x\right)\left(\mathrm{mod}\;2n\right)$. Then, x can be recovered by brute force searching in time $O\left(\left(2\rho q^{t}\right)\left(q^{t}{\left(2\rho \right)}^{3}\right)\right)=O\left(q^{8t}{\left(\log n\right)}^{4}\right)$. In fact, there are $|L_{j_{0}}|\cdot q^{t}$ candidate sequences for x, and for each candidate sequence $x^{\prime }$, we need to verify whether $h_{\mathrm{syn}}\left(x_{L_{j_{0}}}^{\prime }\right)=h_{\mathrm{syn}}\left(x_{L_{j_{0}}}\right)$, which takes time $O(q^{t}|L_{j_{0}}{|}^{3})$ by Lemma 2. Hence, the time of brute force searching is$$O\left(\left(\right|L_{j_{0}}|\cdot q^{t})(q^{t}|L_{j_{0}}{|}^{3})\right)\leq O\left(\left(2\rho q^{t}\right)\left(q^{t}{\left(2\rho \right)}^{3}\right)\right),$$
where the inequality holds because $|L_{j_{0}}|\leq 2\rho$. Thus, the decoding complexity of $C_{\mathrm{syn}}$ is O(nlogn). □

## 5. Correcting Burst-Deletion with Two Reads

In this section, we construct a family of codes correcting a burst of at most *t* deletions with 2 reads. Our construction improves the construction in [31] in redundancy. We first recall the concept of period of a sequence.

Suppose T≤m are two positive integers and $x\in \Sigma_{q}^{m}$. We say that *T* is a *period* of x if $x_{i}=x_{i+T}$ for any i∈[1,m−T].

The following two simple observations are easy to see and will be used in our construction.

Observation 1: Let $x,x^{\prime }\in \Sigma_{q}^{n}$ and $t^{\prime }\in \left[t\right]$. Suppose $D=\left[i_{d},i_{d}+t^{\prime }-1\right],D^{\prime }=\left[i_{d}^{\prime },i_{d}^{\prime }+t^{\prime }-1\right]\subseteq \left[n\right]$ such that $i_{d}\leq i_{d}^{\prime }$. Then, $x_{[n]\setminus D}=x_{\left[n\right]\setminus D^{\prime }}^{\prime }$ if and only if the following holds:$$\begin{array}{c} x_{i}^{\prime }=\left\{\;\begin{array}{c} x_{i},\;\;\;\;\;\mathrm{for}\;i\in \left[1,i_{d}-1\right],\\ x_{i+t^{\prime }},\;\;\;\mathrm{for}\;i\in \left[i_{d},i_{d}^{\prime }-1\right],\\ x_{i},\;\;\;\;\;\mathrm{for}\;i\in \left[i_{d}^{\prime }+t^{\prime },n\right]. \end{array}\right. \end{array}$$

Observation 2: Let $x\in \Sigma_{q}^{n}$ and $t^{\prime }\in \left[t\right]$. Suppose J⊆[n] is an interval such that $\left|J\right|\geq t^{\prime }$ and the substring $x_{J}$ of x has period $t^{\prime }$. Then, for any $D=[i_{d},i_{d}+t^{\prime }-1]\subseteq J$ and $D^{\prime }=\left[i_{d}^{\prime },i_{d}^{\prime }+t^{\prime }-1\right]\subseteq J$, we have $x_{[n]\setminus D}=x_{\left[n\right]\setminus D^{\prime }}$.

Our construction will use (p,δ)-dense sequences for $p=0^{t+1}1^{t+1}$. To apply Lemma 6 to construct (p,δ)-dense sequences, we need to let $\delta =2\left(t+1\right)q^{2(t+1)}\left\lceil \log n\right\rceil$. Then, by Lemma 6, there exists an invertible function $\mathrm{EncDen}:\Sigma_{q}^{n-1}\to \Sigma_{q}^{n}$, such that x=EncDen(u) is (p,δ)-dense for every $u\in \Sigma_{q}^{n-1}$. Moreover, both EncDen and its inverse DecDen are computable in O(nlogn) time. Thus, in this section, we always assume that$$p=0^{t+1}1^{t+1}\;\;\mathrm{and}\;\;\delta =2\left(t+1\right)q^{2(t+1)}\left\lceil \log n\right\rceil .$$

Suppose $x\in \Sigma_{q}^{n}$ is (p,δ)-dense and J⊆[n] is an interval of length |J|≥δ. By Definition 1, there exists an $i_{0}\in J$ such that $J_{0}≜\left[i_{0},i_{0}+2\left(t+1\right)-1\right]\subseteq J$ and $x_{J_{0}}=p=0^{t+1}1^{t+1}$. For any positive integer T≤2t, let $i_{0}^{\prime }=i_{0}+t+1-T$ if T≤t, and let $i_{0}^{\prime }=i_{0}$ if T≥t+1. Then, $\left[i_{0}^{\prime },i_{0}^{\prime }+T\right]\subseteq J_{0}\subseteq J$ and $x_{i_{0}^{\prime }}=0\neq 1=x_{i_{0}^{\prime }+T}$. Thus, *T* is not a period of $x_{J}$. In other words, we have the following remark.

**Remark** **4.**
*Suppose $x\in \Sigma_{q}^{n}$ is (p,δ)-dense, where $p=0^{t+1}1^{t+1}$ and $\delta =2\left(t+1\right)q^{2(t+1)}\left\lceil \log n\right\rceil$. Then, the length of any substring of x of period T≤2t is at most δ.*


**Lemma** **9.**
*Suppose $x\neq x^{\prime }\in \Sigma_{q}^{n}$ are both (p,δ)-dense. If $|B_{\leq t}\left(x\right)\cap B_{\leq t}\left(x^{\prime }\right)|\geq 2$, then there exists an interval J⊆[n] of length |J|≤δ+t and two intervals $D,D^{\prime }\subseteq J$ of size $|D|=|D^{\prime }|\leq t$, such that $x_{[n]\setminus D}=x_{\left[n\right]\setminus D^{\prime }}^{\prime }$.*


**Proof.** Suppose $y,y^{\prime }\in B_{\leq t}\left(x\right)\cap B_{\leq t}\left(x^{\prime }\right)$ and $y\neq y^{\prime }$. Then, $y=x_{\left[n\right]\setminus D_{1}}=x_{\left[n\right]\setminus D_{1}^{\prime }}^{\prime }$ for some intervals $D_{1},D_{1}^{\prime }\subseteq \left[n\right]$ of size $t_{1}≜|D_{1}|=|D_{1}^{\prime }|\leq t$, and $y^{\prime }=x_{\left[n\right]\setminus D_{2}}=x_{\left[n\right]\setminus D_{2}^{\prime }}^{\prime }$ for some intervals $D_{2},D_{2}^{\prime }\subseteq \left[n\right]$ of size $t_{2}≜|D_{2}|=|D_{2}^{\prime }|\leq t$. For convenience in the discussions, we denote$$D_{j}=\left[i_{j},i_{j}+t_{j}-1\right]\;\mathrm{and}\;D_{j}^{\prime }=\left[i_{j}^{\prime },i_{j}^{\prime }+t_{j}-1\right],\;j=1,2.$$If $|i_{j}-i_{j}^{\prime }|\leq \delta$ for some j∈{1,2}, then let $D=D_{j}$, $D^{\prime }=D_{j}^{\prime }$ and $J=[\lambda ,\lambda^{\prime }+t_{j}-1]$ such that $\lambda =\min\{i_{j},i_{j}^{\prime }\}$ and $\lambda^{\prime }=\max\left\{i_{j},i_{j}^{\prime }\right\}$. Clearly, *J*, *D*, and $D^{\prime }$ satisfy the desired conditions. So, in the following, we assume that $|i_{j}-i_{j}^{\prime }|>\delta$ for each j∈{1,2}.By the symmetry of x and $x^{\prime }\;(y$ and $y^{\prime })$, we can assume$$i_{1}<i_{1}^{\prime }\;\;\mathrm{and}\;\;t_{2}\leq t_{1}.$$Since $y=x_{\left[n\right]\setminus D_{1}}=x_{\left[n\right]\setminus D_{1}^{\prime }}^{\prime }$, where $D_{1}=\left[i_{1},i_{1}+t_{1}-1\right]$, $D_{1}^{\prime }=\left[i_{1}^{\prime },i_{1}^{\prime }+t_{1}-1\right]$ and $i_{1}<i_{1}^{\prime }$, we can obtain from Observation 1 that(20)$$\begin{array}{c} x_{i}^{\prime }=\left\{\;\begin{array}{c} y_{i}=x_{i},\;\;\;\;\;\mathrm{for}\;i\in \left[1,i_{1}-1\right],\\ y_{i}=x_{i+t_{1}},\;\;\;\mathrm{for}\;i\in \left[i_{1},i_{1}^{\prime }-1\right],\\ y_{i-t_{1}}=x_{i},\;\;\;\mathrm{for}\;i\in \left[i_{1}^{\prime }+t_{1},n\right]. \end{array}\right. \end{array}$$Similarly, since $y^{\prime }=x_{\left[n\right]\setminus D_{2}}=x_{\left[n\right]\setminus D_{2}^{\prime }}^{\prime }$, where $D_{2}=\left[i_{2},i_{2}+t_{2}-1\right]$ and $D_{2}^{\prime }=\left[i_{2}^{\prime },i_{2}^{\prime }+t_{2}-1\right]$, we have
If $i_{2}<i_{2}^{\prime }$, then according to Observation 1,(21)$$\begin{array}{c} x_{i}^{\prime }=\left\{\;\begin{array}{c} y_{i}^{\prime }=x_{i},\;\;\;\;\;\mathrm{for}\;i\in \left[1,i_{2}-1\right],\\ y_{i}^{\prime }=x_{i+t_{2}},\;\;\;\mathrm{for}\;i\in \left[i_{2},i_{2}^{\prime }-1\right],\\ y_{i-t_{2}}^{\prime }=x_{i},\;\;\;\mathrm{for}\;i\in \left[i_{2}^{\prime }+t_{2},n\right]. \end{array}\right. \end{array}$$If $i_{2}>i_{2}^{\prime }$, then according to Observation 1,(22)$$\begin{array}{c} x_{i}^{\prime }=\left\{\;\begin{array}{c} y_{i}^{\prime }=x_{i},\;\;\;\;\;\;\;\;\;\mathrm{for}\;i\in \left[1,i_{2}^{\prime }-1\right],\\ y_{i-t_{2}}^{\prime }=x_{i-t_{2}},\;\;\;\mathrm{for}\;i\in \left[i_{2}^{\prime }+t_{2},i_{2}+t_{2}-1\right],\\ y_{i}^{\prime }=x_{i},\;\;\;\;\;\;\;\;\;\mathrm{for}\;i\in \left[i_{2}+t_{2},n\right]. \end{array}\right. \end{array}$$Let$$I_{1}=\left[i_{1},i_{1}^{\prime }+t_{1}-1\right]\;\;\mathrm{and}\;\;I_{2}=\left[{\bar{i}}_{2},{\bar{i}}_{2}^{\prime }+t_{2}-1\right],$$
where ${\bar{i}}_{2}=\min\left\{i_{2},i_{2}^{\prime }\right\}$ and ${\bar{i}}_{2}^{\prime }=\max\left\{i_{2},i_{2}^{\prime }\right\}$. Then, by (Equation 20), (Equation 21), and (Equation 22), we can easily obtain the following claim.Claim 1: For all $i\in \left[n\right]\setminus \left(I_{1}\cap I_{2}\right)$, we have $x_{i}=x_{i}^{\prime }$.Since $x\neq x^{\prime }$, by Claim 1, we have $I_{1}\cap I_{2}\neq Ø$. If $|I_{1}\cap I_{2}|\leq t$ then, clearly, $J=D=D^{\prime }=I_{1}\cap I_{2}$ satisfies the desired conditions. So, in the following, we only need to consider $|I_{1}\cap I_{2}|>t$. We have the following two cases.Case 1: $i_{2}<i_{2}^{\prime }$. Then, $\min\left\{i_{2},i_{2}^{\prime }\right\}=i_{2}$ and $\max\left\{i_{2},i_{2}^{\prime }\right\}=i_{2}^{\prime }$, and so $I_{2}=\left[i_{2},i_{2}^{\prime }+t_{2}-1\right]$ and $I_{1}\cap I_{2}=\left[\lambda ,\lambda^{\prime }\right]$, where $\lambda =\max\{i_{1},i_{2}\}$ and $\lambda^{\prime }=\min\left\{i_{1}^{\prime }+t_{1}-1,i_{2}^{\prime }+t_{2}-1\right\}$. We need to further divide this case into the following two subcases.Case 1.1: $t_{2}<t_{1}$. Let $D=[\lambda ,\lambda +t_{2}-1]$ and $D^{\prime }=\left[\lambda^{\prime }-t_{2}+1,\lambda^{\prime }\right]$. Then, $D,D^{\prime }\subseteq J$ and $|D|=|D^{\prime }|\leq t$. Note that by Claim 1, $x_{i}^{\prime }=x_{i}$ for every $i\in \left[n\right]\setminus \left(I_{1}\cap I_{2}\right)=\left[n\right]\setminus \left[\lambda ,\lambda^{\prime }\right]$, and by (Equation 20), $x_{i}^{\prime }=x_{i+t_{2}}$ for every $i\in [\lambda ,\lambda^{\prime }-t_{2}]$. So, according to Observation 1, we can obtain$$x_{[n]\setminus D}=x_{\left[n\right]\setminus D^{\prime }}^{\prime }.$$Moreover, by (Equation 20) and (Equation 21), $x_{i}=x_{i-t_{2}}^{\prime }=x_{i-t_{2}+t_{1}}$ for every $i\in [\lambda +t_{2},\lambda^{\prime }-t_{1}+t_{2}]$. So, $x_{\left[\lambda +t_{2},\lambda^{\prime }\right]}$ has period $t_{1}-t_{2}$, and by Remark 4, $|[\lambda +t_{2},\lambda^{\prime }]|\leq \delta$. Hence, we have$$|J|=|I_{1}\cap I_{2}|=|[\lambda ,\lambda^{\prime }]|\leq \delta +t_{2}\leq \delta +t.$$Case 1.2: $t_{2}=t_{1}$. We will prove that this case is impossible by contradiction. Without loss of generality, assume $i_{1}\leq i_{2}$. Since $|I_{1}\cap I_{2}|>t$, then $I_{1}\cap I_{2}=\left[i_{2},{\tilde{i}}_{1}^{\prime }\right]$ and $i_{2}\leq {\tilde{i}}_{1}^{\prime }-t$, where ${\tilde{i}}_{1}^{\prime }=\min\left\{i_{1}^{\prime },i_{2}^{\prime }\right\}$. By (Equation 20) and (Equation 21), we have$$x_{i}=x_{i}^{\prime }=x_{i+t_{1}}$$
for every $i\in [i_{1},i_{2}-1]$, i.e., $x_{\left[i_{1},i_{2}+t_{1}-1\right]}$ has period $t_{1}$. So, we can obtain $y=x_{\left[n\right]\setminus \left[i_{1},i_{1}+t_{1}-1\right]}=x_{\left[n\right]\setminus \left[i_{2},i_{2}+t_{1}-1\right]}=y^{\prime }$ (according to Observation 2), which contradicts to the assumption that $y\neq y^{\prime }$.Case 2: $i_{2}>i_{2}^{\prime }$. In this case, $I_{2}=\left[i_{2}^{\prime },i_{2}+t_{2}-1\right]$ and we have $I_{1}\cap I_{2}=\left[\lambda ,\lambda^{\prime }\right]$, where $\lambda =\max\{i_{1},i_{2}^{\prime }\}$ and $\lambda^{\prime }=\min\left\{i_{1}^{\prime }+t_{1}-1,i_{2}+t_{2}-1\right\}$. Let $D=[\lambda ,\lambda +t_{1}-1]$ and $D^{\prime }=\left[\lambda^{\prime }-t_{1}+1,\lambda^{\prime }\right]$. Then, $D,D^{\prime }\subseteq J$ and $|D|=|D^{\prime }|\leq t$. Note that by Claim 1, $x_{i}^{\prime }=x_{i}$ for every $i\in \left[n\right]\setminus \left(I_{1}\cap I_{2}\right)=\left[n\right]\setminus \left[\lambda ,\lambda^{\prime }\right]$, and by (Equation 20), $x_{i}^{\prime }=x_{i+t_{1}}$ for every $i\in [\lambda ,\lambda^{\prime }-t_{1}]$. So, according to Observation 1, we can obtain$$x_{[n]\setminus D}=x_{\left[n\right]\setminus D^{\prime }}^{\prime }.$$Moreover, by (Equation 20) and (Equation 22), we can obtain$$x_{i}=x_{i+t_{2}}^{\prime }=x_{i+t_{2}+t_{1}}$$
for every $i\in [\lambda ,\lambda^{\prime }-t_{1}-t_{2}]$. Hence, $x_{\left[\lambda ,\lambda^{\prime }\right]}$ is a substring of x of period $t_{1}+t_{2}$. By Remark 4, $|[\lambda ,\lambda^{\prime }]|\leq \delta$, so $|J|=|I_{1}\cap I_{2}|=|[\lambda ,\lambda^{\prime }]|\leq \delta \leq \delta +t$. □

For n=30, we give three examples of y and $y^{\prime }$, satisfying conditions of the different cases in the proof of Lemma 9.

**Example** **1.**
*Suppose $y=x_{\left[n\right]\setminus D_{1}}=x_{\left[n\right]\setminus D_{1}^{\prime }}^{\prime }$ and $y^{\prime }=x_{\left[n\right]\setminus D_{2}}=x_{\left[n\right]\setminus D_{2}^{\prime }}^{\prime }$, where $D_{1}=\left\{4,5,6,7\right\}$, $D_{1}^{\prime }=\left\{27,28,29,30\right\}$, $D_{2}=\left\{1,2\right\}$ and $D_{2}^{\prime }=\left\{23,24\right\}$. Here, we have $i_{1}=4,i_{1}^{\prime }=27$ and $t_{1}=4$; $i_{2}=1,i_{2}^{\prime }=23$ and $t_{2}=2$. Then, conditions of Case 1.1 are satisfied, and $J=I_{1}\cap I_{2}=\left[4,24\right]$. Figure 1a is an illustration of this case. We can easily find that $x_{i}^{\prime }=x_{i+2}$ for every i∈[4,22] and $x_{i}^{\prime }=x_{i}$ for every i∈[n]∖J. So, by Observation 1, $x_{\left[n]\setminus \{4,5\right\}}=x_{\left[n]\setminus \{23,24\right\}}^{\prime }$. Moreover, $x_{i}=x_{i-2}^{\prime }=x_{i+2}$ for every i∈[6,22], so the substring $x_{\left[6,24\right]}$ of x has period 2.*


**Example** **2.**
*Suppose $y=x_{\left[n\right]\setminus D_{1}}=x_{\left[n\right]\setminus D_{1}^{\prime }}^{\prime }$ and $y^{\prime }=x_{\left[n\right]\setminus D_{2}}=x_{\left[n\right]\setminus D_{2}^{\prime }}^{\prime }$, where $D_{1}=\left\{1,2,3,4\right\}$, $D_{1}^{\prime }=\left\{17,18,19,20\right\}$, $D_{2}=\left\{9,10,11,12\right\}$ and $D_{2}^{\prime }=\left\{27,28,29,30\right\}$. Here, we have $i_{1}=1,i_{1}^{\prime }=17$, $i_{2}=9,i_{2}^{\prime }=27$ and $t_{1}=t_{2}=4$. Then, conditions of Case 1.2 are satisfied. Figure 1b is an illustration of this case. We can find that $x_{i}=x_{i}^{\prime }=x_{i+4}$ for every i∈[1,8], i.e., $x_{\left[1,12\right]}$ has period 4. So, by Observation 2, we have $y=x_{\left[n\right]\setminus D_{1}}=x_{\left[n\right]\setminus D_{2}}=y^{\prime }$.*


**Example** **3.**
*Suppose $y=x_{\left[n\right]\setminus D_{1}}=x_{\left[n\right]\setminus D_{1}^{\prime }}^{\prime }$ and $y^{\prime }=x_{\left[n\right]\setminus D_{2}}=x_{\left[n\right]\setminus D_{2}^{\prime }}^{\prime }$, where $D_{1}=\left\{7,8,9,10\right\}$, $D_{1}^{\prime }=\left\{27,28,29,30\right\}$, $D_{2}=\left\{24,25,26\right\}$ and $D_{2}^{\prime }=\left\{1,2,3\right\}$. Here, we have $i_{1}=7,i_{1}^{\prime }=27$ and $t_{1}=4$; $i_{2}=24,i_{2}^{\prime }=1$ and $t_{2}=3$. Then, conditions of Case 2 are satisfied and $J=I_{1}\cap I_{2}=\left[7,26\right]$. Figure 1c is an illustration of this case. We can easily find that $x_{i}^{\prime }=x_{i+4}$ for every i∈[7,22], and $x_{i}^{\prime }=x_{i}$ for every i∈[n]∖J. So by Observation 1, $x_{\left[n]\setminus \{7,8,9,10\right\}}=x_{\left[n]\setminus \{23,24,25,26\right\}}^{\prime }$. Moreover, $x_{i}=x_{i+3}^{\prime }=x_{i+7}$ for every i∈[7,19], i.e., $x_{\left[7,26\right]}$ has period 7.*


In the following, we give a construction of *q*-ary codes for correcting a burst of at most *t* deletions with two reads. Let$$\rho =\delta +t=2\left(t+1\right)q^{2(t+1)}\left\lceil \log n\right\rceil +t$$
and $L_{\rho }=\left\{L_{j}:j=1,2,\cdots ,\left\lceil n/\rho \right\rceil -1\right\}$ be defined by (Equation 2). For each ℓ∈{0,1}, let the function ${\tilde{h}}_{\rho }^{\left(\ell \right)}$ be obtained from Construction 1 by letting *h* be the function $h_{\mathrm{syn}}$ given by Lemma 2. Then, let(23)$$\begin{array}{c} \tilde{f}\left(x\right)=\left({\tilde{h}}_{\rho }^{\left(0\right)}\left(x\right),{\tilde{h}}_{\rho }^{\left(1\right)}\left(x\right)\right). \end{array}$$

**Lemma** **10.**
*For each $x\in \Sigma_{q}^{n}$, the function $\tilde{f}\left(x\right)$ is computable in time $O(q^{7t}n{\left(\log n\right)}^{3})$, and when viewed as a binary string, the length $\left|\tilde{f}\left(x\right)\right|$ of $\tilde{f}\left(x\right)$ satisfies*

$$|\tilde{f}\left(x\right)|\leq 8\log\log n+o\left(\log\log n\right).$$


*Moreover, if x is (p,δ)-dense, then given $\tilde{f}\left(x\right)$ and any two distinct $y,y^{\prime }\in B_{\leq t}\left(x\right)$, one can uniquely recover x.*


**Proof.** Note that $|L_{j}|\leq 2\rho =4\left(t+1\right)q^{2(t+1)}\left\lceil \log n\right\rceil +2t$ for each *j*. Then, by (Equation 3) and by Remark 2, the functions ${\tilde{h}}_{\rho }^{\left(0\right)}\left(x\right)$ and ${\tilde{h}}_{\rho }^{\left(1\right)}\left(x\right)$ are computable in time$$O\left(nq^{t}{\left(2\rho \right)}^{3}\right)=O\left(q^{7t}n{\left(\log n\right)}^{3}\right).$$Hence, by (Equation 23), $\tilde{f}\left(x\right)$ is computable in time $O(q^{7t}n{\left(\log n\right)}^{3})$.Again by (Equation 3) and by Remark 2, the length of $\tilde{f}\left(x\right)$ (viewed as a binary string) satisfies$$\begin{array}{cc} \left|\tilde{f}\left(x\right)\right| & =\;|{\tilde{h}}_{\rho }^{\left(0\right)}\left(x\right)|+|{\tilde{h}}_{\rho }^{\left(1\right)}\left(x\right)|\\ & =2\log q^{R_{2\rho }}\\ & =2\log q^{4\log_{q}\left(2\rho \right)+o\left(\log_{q}\left(2\rho \right)\right)}\\ & =8\log\log n+o(\log\log n), \end{array}$$
where the last equality comes from the assumption that $\rho =\delta +t=2\left(t+1\right)q^{2(t+1)}\left\lceil \log n\right\rceil +t$.Finally, we prove that if $x\in \Sigma_{q}^{n}$ is (p,δ)-dense, then given $\tilde{f}\left(x\right)$ and any two distinct $y,y^{\prime }\in B_{\leq t}\left(x\right)$, one can uniquely recover x. It suffices to prove that for any (p,δ)-dense $x,x^{\prime }\in \Sigma_{q}^{n}$, if $|B_{\leq t}(x)\cap B_{\leq t}(x^{\prime })|\geq 2$, then $\tilde{f}\left(x\right)\neq \tilde{f}\left(x^{\prime }\right)$. This can be proved as follows. By Lemma 9, there exists an interval J⊆[n] of length |J|≤δ+t=ρ and two intervals $D,D^{\prime }\subseteq J$ of size $\left|D|=\right|D^{\prime }|\leq t$, such that $x_{[n]\setminus D}=x_{\left[n\right]\setminus D^{\prime }}^{\prime }$. Then, by Lemma 3, we have ${\tilde{h}}_{\rho }^{\left(\ell \right)}\left(x\right)\neq {\tilde{h}}_{\rho }^{\left(\ell \right)}\left(x^{\prime }\right)$ for some ℓ∈{0,1}. Therefore, by (Equation 23), we have $\tilde{f}\left(x\right)\neq \tilde{f}\left(x^{\prime }\right)$. □

**Theorem** **2.**
*Let ${\tilde{C}}_{\mathrm{syn}}$ be the code with the encoding function*

(24)
$$\begin{array}{cc} \tilde{E}:\;\Sigma_{q}^{n-1} & \;\to \;\;\Sigma_{q}^{n+r}\\ u\;\; & \;\mapsto \;\left(x,w\right) \end{array}$$

*where x=EncDen(u), $w=\left(0^{t}1,x_{\left[n-t+1,n\right]},{\tilde{f}}_{q}\left(x\right)\right)$, and $r=|w|=2t\;+\;1\;+\;|{\tilde{f}}_{q}\left(x\right)|$. Then, ${\tilde{C}}_{\mathrm{syn}}$ is an $\left(n,2,B_{\leq t}\right)$-reconstruction code with redundancy 8loglogn+o(loglogn) bits, encoding complexity $O\left(q^{7t}n{\left(\log n\right)}^{3}\right)$ and decoding complexity $O\left(q^{9t}{\left(n\log n\right)}^{3}\right)$.*


**Proof.** We first prove that ${\tilde{C}}_{\mathrm{syn}}$ is an $\left(n,2,B_{\leq t}\right)$-reconstruction code. We need to prove that for each codeword $z=\tilde{E}\left(u\right)=\left(x,0^{t}1,x_{\left[n-t+1,n\right]},{\tilde{f}}_{q}\left(x\right)\right)\in {\tilde{C}}_{\mathrm{syn}}$, given any distinct $y,y^{\prime }\in B_{\leq t}\left(z\right)$, one can uniquely recover x.Let $y=z_{[n+r]\setminus D}$, where D⊆[n+r] is an interval of size $t^{\prime }=\left|D\right|=\left|z\right|-\left|y\right|$. By the same discussions as in the proof of Lemma 8, exactly one of the following three cases holds:Case 1: D⊆[n+1,n+r]. In this case, we have $x=y_{\left[1,n\right]}$.Case 2: There exists a $t^{\prime \prime }\in \left[t^{\prime }-1\right]$ such that $D\subseteq [n-t^{\prime }+t^{\prime \prime }+1,n+t^{\prime \prime }]$. In this case, $y_{\left[1,n+1-t^{\prime }+t^{\prime \prime }\right]}=x_{\left[1,n+1-t^{\prime }+t^{\prime \prime }\right]}$ and $y_{\left[n+t+2-t^{\prime },n+2t+1-t^{\prime }\right]}=z_{\left[n+t+2,n+2t+1\right]}=x_{\left[n-t+1,n\right]}$. So, $x=(y_{\left[1,n+1-t\right]},y_{\left[n+t+2-t^{\prime },n+2t+1-t^{\prime }\right]})$.Case 3: D⊆[n]. In this case, we have $y_{\left[n-t^{\prime }\right]}\in B_{\leq t}\left(x\right)$ and $y_{\left[n+2t+1-t^{\prime },n+r-t^{\prime }\right]}={\tilde{f}}_{q}\left(x\right)$.Similarly, for $y^{\prime }\in B_{\leq t}\left(z\right)$, either x can be recovered from $y^{\prime }$ or $y^{\prime }=z_{\left[n+r\right]\setminus D^{\prime }}$ for some interval $D^{\prime }\subseteq \left[n\right]$ of size $|D^{\prime }|=|z|-|y^{\prime }|\leq t$. Suppose $y=z_{[n+r]\setminus D}$ and $y^{\prime }=z_{\left[n+r\right]\setminus D^{\prime }}$ for some intervals $D,D^{\prime }\subseteq \left[n\right]$. Then, $y_{\left[n-|D|\right]},y_{\left[n-|D^{\prime }|\right]}^{\prime }\in B_{\leq t}\left(x\right)$ and $y_{\left[n+2t+1-|D^{\prime }|,n+r-|D^{\prime }|\right]}^{\prime }={\tilde{f}}_{q}\left(x\right)$. Moreover, we must have $y_{\left[n-|D|\right]}\neq y_{\left[n-|D^{\prime }|\right]}^{\prime }$, because otherwise, from (Equation 24), we will obtain $y=\left(y_{\left[n-|D|\right]},w\right)=\left(y_{\left[n-|D^{\prime }|\right]}^{\prime },w\right)=y^{\prime }$, which contradicts to the assumption that $y\neq y^{\prime }$. By Lemma 10, x can be uniquely recovered from y and $y^{\prime }$.Thus, we proved that, given any distinct $y,y^{\prime }\in B_{\leq t}\left(z\right)$, one can uniquely recover x (and so z), which implies that ${\tilde{C}}_{\mathrm{syn}}$ is an $\left(n,2,B_{\leq t}\right)$-reconstruction code.By (Equation 24), the redundancy of ${\tilde{C}}_{\mathrm{syn}}$ is $r+1=t+2+|{\tilde{f}}_{q}\left(x\right)|$ in *q*-ary symbols, which is at most 8loglogn+o(loglogn) bits by Lemma 10.The encoding complexity of ${\tilde{C}}_{\mathrm{syn}}$ comes from the complexity of computing ${\tilde{f}}_{q}\left(x\right)$, which is $O\left(q^{7t}n{\left(\log n\right)}^{3}\right)$ by Lemma 10. The decoding complexity of ${\tilde{C}}_{\mathrm{syn}}$ by brute force searching is at most $O\left({\left(nq^{t}\right)}^{2}q^{7t}n{\left(\log n\right)}^{3}\right)=O\left(q^{9t}{\left(n\log n\right)}^{3}\right)$. □

## 6. Conclusions and Discussions

We proposed new constructions of *q*-ary codes correcting a burst of at most *t* deletions, for both classical error correcting codes and reconstruction codes. By using *q*-ary (p,δ)-dense strings and bounded burst-deletion correcting codes, our constructions reduce the code redundancy in the loglogn term and have simpler encoding functions, compared to existing works.

A more general problem is to construct *q*-ary $\left(n,N,D_{t}\right)$-reconstruction codes, i.e., *q*-ary reconstruction codes of length *n* for *t*-deletion channel with *N* reads. Note that the best known explicit construction (regarding the redundancy) for classical *t*-deletion correcting binary codes of length *n* has redundancy (4t−1)logn+o(logn) [11]. So, we are interested in constructing *q*-ary $\left(n,N,D_{t}\right)$-reconstruction codes, for any fixed *q* and *t*, of length *n* and with redundancy klogn+o(logn) bits for some positive integer *k* such that 1≤k<4t−1 and *N* depends only on *k* and *t* (and is independent of *n*).

## Figures and Tables

**Figure 1 entropy-27-00085-f001:**
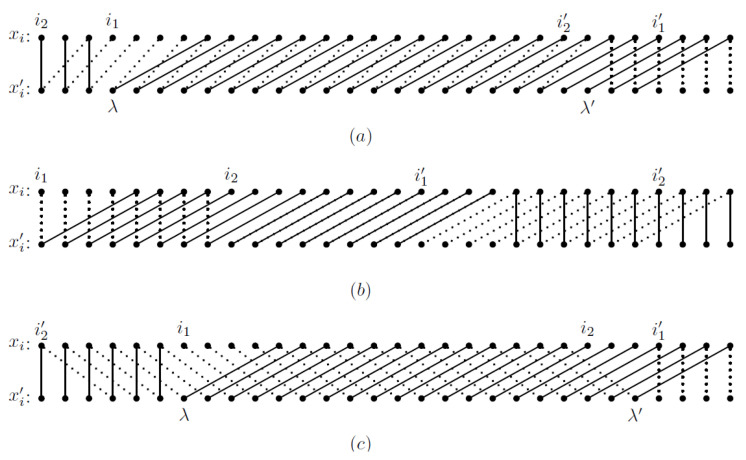
Illustration examples: each dot in the upper row represents a symbol of x, and each dot in the lower row represents a symbol of $x^{\prime }$, where two symbols with equal value are connected by a (solid or dashed) line segment. We can find that: (1) in (**a**), the substring $x_{\left[6,24\right]}$ of x has period 2; (2) in (**b**), the substring $x_{\left[1,12\right]}$ of x has period 4, so $x_{\left[n]\setminus \{1,2,3,4\right\}}=x_{\left[n]\setminus \{9,10,11,12\right\}}$; and (3) in (**c**), the substring $x_{\left[7,26\right]}$ of x has period 7.

## Data Availability

No new data were created or analyzed in this study. Data sharing is not applicable to this article.

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
