# Peer review of "Some New Constructions of *q*-ary Codes for Correcting a Burst of at Most *t* Deletions [Author-notes fn1-entropy-27-00085]"

_entropy, 2025, doi:10.3390/e27010085_

Round 1
Reviewer 1 Report
Comments and Suggestions for Authors
A strange problem of finding codes for correcting burst deletions from two reads requires more motivation. In classic coding theory, they usually treats "two reads" as a repetition code.
The coding procedure in Lemma 2 is far from optimal.
Lemma 6 needs some extra conditions since it is not true if n and \delta are not large enough.
Author Response
Comments 1: A strange problem of finding codes for correcting burst deletions from two reads requires more motivation. In classic coding theory, they usually treats "two reads" as a repetition code.
Response: Thank you for the valuable comments. We have added a centence in line 62 as follows: This model is suitable for DNA data storage because current synthesis and sequencing technologies can generate many (possibly erroneous) reads for each DNA strand, and so each stored DNA strand can be recovered by its many erroneous copies.
Comments 2: The coding procedure in Lemma 2 is far from optimal.
Response: Yes. The construction in Lemma 2 is not our result, it is only an auxiliary tool, which is used in our new construction.
Comments 3: Lemma 6 needs some extra conditions since it is not true if n and \delta are not large enough.
Response: Yes. We assume sufficiently large n in line 308 by the sentence ``As we are interested in large n, we will always assume that ...''. In the same paragraph, we also assume the value of \delta.
Reviewer 2 Report
Comments and Suggestions for Authors
This paper has introduced new constructions of q-ary codes that correct a burst of at most t deletions. It has applications to DNA storage and some nonvolatile memory channels. The work considers two paradigms for error correction: classical error correction, and reconstruction codes (where the codeword can be read multiple times to get multiple copies with different errors). For both cases, the paper has constructed codes of state-of-the-art performance: their redundancy is lower than the best known existing works. The paper also gives explicit encoding functions for both constructions that are simpler than previous works.
Overall, the paper contains interesting new work, and is very well presented.
Author Response
Thank you very much for your positive comments.